# Target-specific control of olfactory bulb periglomerular cells by GABAergic and cholinergic basal forebrain inputs

**Didier De Saint Jan***

Institut des Neurosciences Cellulaires et Intégratives, Centre National de la Recherche Scientifique, Strasbourg, France

**Abstract** The olfactory bulb (OB), the first relay for odor processing in the brain, receives dense GABAergic and cholinergic long-range projections from basal forebrain (BF) nuclei that provide information about the internal state and behavioral context of the animal. However, the targets, impact, and dynamic of these afferents are still unclear. How BF synaptic inputs modulate activity in diverse subtypes of periglomerular (PG) interneurons using optogenetic stimulation and loose cell-attached or whole-cell patch-clamp recording in OB slices from adult mice were studied in this article. GABAergic BF inputs potently blocked PG cells firing except in a minority of calretinin-expressing cells in which GABA release elicited spiking. Parallel cholinergic projections excited a previously overlooked PG cell subtype via synaptic activation of M1 muscarinic receptors. Low-frequency stimulation of the cholinergic axons drove persistent firing in these PG cells, thereby increasing tonic inhibition in principal neurons. Taken together, these findings suggest that modality-specific BF inputs can orchestrate synaptic inhibition in OB glomeruli using multiple, potentially independent, inhibitory or excitatory target-specific pathways.

## Editor's evaluation

This study reports on the synaptic impact of basal forebrain stimulation on a population of olfactory bulb interneurons in acute mouse brain slices. The author reveals that optogenetic stimulation of GABAergic basal forebrain afferents by and large inhibits the discharge of periglomerular cells, whereas cholinergic afferents evoke a prolonged, M1 receptor-mediated depolarization and increase in firing in a subpopulation of periglomerular cells. The current study would further our understanding of the olfactory neural circuit and how different neurotransmitters shape postsynaptic neuronal responses.

## Introduction

Basal forebrain (BF) nuclei innervate many regions of the brain, including the entire neocortex, hippocampus, amygdala, thalamus, and hypothalamus, with diffuse long-range projections releasing GABA, ACh, and, more rarely, glutamate. These projections provide cues about the behavioral context and internal state of the animal. They modulate multiple synaptic, cellular, and network processes at a variety of temporal and spatial scales, thereby regulating sensory perception, metabolic functions such as food intake, brain states, and important cognitive functions, including attention, arousal, memory or learning (*Ballinger et al., 2016*; *Picciotto et al., 2012*).

The olfactory bulb (OB), the first region that processes olfactory information in the brain, receives massive cholinergic and GABAergic BF projections that principally originate in the nucleus of the horizontal limb of the diagonal band of Broca (HDB) and in the magnocellular preoptic nucleus

**\*For correspondence:**
desaintjan@inci-cnrs.unistra.fr

**Competing interest:** The author declares that no competing interests exist.

(MCPO) (*Záborszky et al., 1986*). Cholinergic signaling within the OB modulates olfactory learning and memory (*Devore et al., 2014*; *Devore et al., 2012*; *Ravel et al., 1994*; *Ross et al., 2019*), odor discrimination (*Chan et al., 2017*; *Chaudhury et al., 2009*; *Doty et al., 1999*; *Li and Cleland, 2013*; *Mandairon et al., 2006*; *Smith et al., 2015*), odor habituation (*Ogg et al., 2018*), and social interactions (*Suyama et al., 2021*). How ACh modulates these behavioral demands is less clear. Cholinergic axons innervate all layers of the OB and preferentially form synapses on inhibitory interneurons (*Hamamoto et al., 2017*; *Kasa et al., 1995*). Yet, a sparse subclass of deep short-axon cells in the internal plexiform layer is the only synaptic target of BF cholinergic axons known to date (*Case et al., 2017*). Deep short-axon cells are few compared to granule cells and periglomerular (PG) cells, the two prevalent classes of GABAergic interneurons in the OB. Moreover, cholinergic synapses are particularly abundant in the glomerular layer (*Hamamoto et al., 2017*), where muscarinic receptors exert a strong control on intraglomerular inhibition (*Liu et al., 2015*), suggesting that PG cells are another likely target of cholinergic axons.

BF GABAergic afferents innervate all layers of the OB at least as densely as cholinergic axons, but only a few studies have examined their function in odor processing (*Böhm et al., 2020*; *Nunez-Parra et al., 2013*). This is a difficult question because BF GABAergic afferents innervate many types of interneurons, including granule and PG cells, which inhibit principal neurons (*Hanson et al., 2020*; *Nunez-Parra et al., 2013*; *Sanz Diez et al., 2019*; *Villar et al., 2021*), as well as deep short-axon cells, which inhibit granule and PG cells (*Case et al., 2017*; *Sanz Diez et al., 2019*). Thus, depending on their target, BF GABAergic inputs may inhibit or disinhibit principal neurons. Moreover, BF GABAergic inputs have target-specific release properties, suggesting that they arise from distinct populations of BF GABAergic neurons (*Sanz Diez et al., 2019*). Finally, the effect on principal neurons may be hard to predict as GABA is excitatory in some PG cells (*Parsa et al., 2015*) and can be co-released along with ACh from cholinergic terminals. Co-transmission of the two transmitters occurs onto deep short-axon cells (*Case et al., 2017*), but it remains unclear whether co-release is systematic in the OB as it is in the hippocampus (*Takács et al., 2018*) or target-specific as in the cortex (*Desikan et al., 2018*; *Saunders et al., 2015*).

In such complex context, an important step towards understanding the influence and function of the BF inputs in the OB is to investigate the connections, temporal dynamics, and functional impact of each BF pathway. Recent papers have examined BF GABAergic inputs onto granule cells and their influence on adult-born granule cells survival (*Hanson et al., 2020*), the firing of principal neurons, and low-field potential rhythmic activity (*Villar et al., 2021*). Here, I examined how synaptic release of GABA and ACh from BF fibers modulates the activity of the various PG cells. PG cells are small axonless GABAergic interneurons with dendritic projections in a single glomerulus. Collectively, they regulate glutamate release from olfactory sensory neurons (OSNs) terminals (*Murphy et al., 2005*; *Shao et al., 2009*), as well as spike timing (*Geramita and Urban, 2017*; *Najac et al., 2015*) and respiration-coupled theta rhythms in output neurons (*Fukunaga et al., 2014*; *Villar et al., 2021*). PG cells are classified into two broad classes with largely similar morphologies but distinct excitatory inputs: type 1 PG cells receive direct excitatory inputs from OSNs, whereas type 2 PG cells are not connected to OSNs and receive glutamatergic inputs from the dendrites of mitral and tufted cells (*Shao et al., 2009*). In our previous study, we have shown that BF GABAergic inputs are an additional criterion that differentiate PG cell subclasses (*Sanz Diez et al., 2019*). This is summarized in *Figure 1*. First, BF GABAergic neurons contact type 2 PG cells but not type 1 PG cells (*Figure 1A*). Second, BF GABAergic inhibitory postsynaptic currents (IPSCs) have target-specific time courses in three classes of type 2 PG cells (*Figure 1B*) that also differ on the basis of their olfactory nerve (ON)-evoked excitatory response, firing properties, and molecular markers (*Figure 1C*). For clarity, I hereafter introduce a new nomenclature and call these three classes type 2.1, type 2.2, and type 2.3 PG cells. Type 2.1 cells correspond to the most abundant calretinin (CR)-expressing PG cells. They conserve intrinsic properties of immature neurons, that is, they have a remarkably high membrane resistance (>1 GΩ, usually higher than in other PG cells), they do not fire or fire at most a single and often small action potential, they receive little excitatory inputs, and their output is uncertain (*Benito et al., 2018*; *Fogli Iseppe et al., 2016*). Type 2.2 cells are labeled in the Kv3.1-eYFP mouse and include calbindin (CB)-positive as well as CB-negative cells. They receive short bursts of excitatory inputs from mitral and tufted cells and, in turn, indiscriminately release GABA onto these principal neurons (*Najac et al., 2015*). Finally, type 2.3 PG cells are a previously ignored subclass of regularly firing PG cells with no known chemical

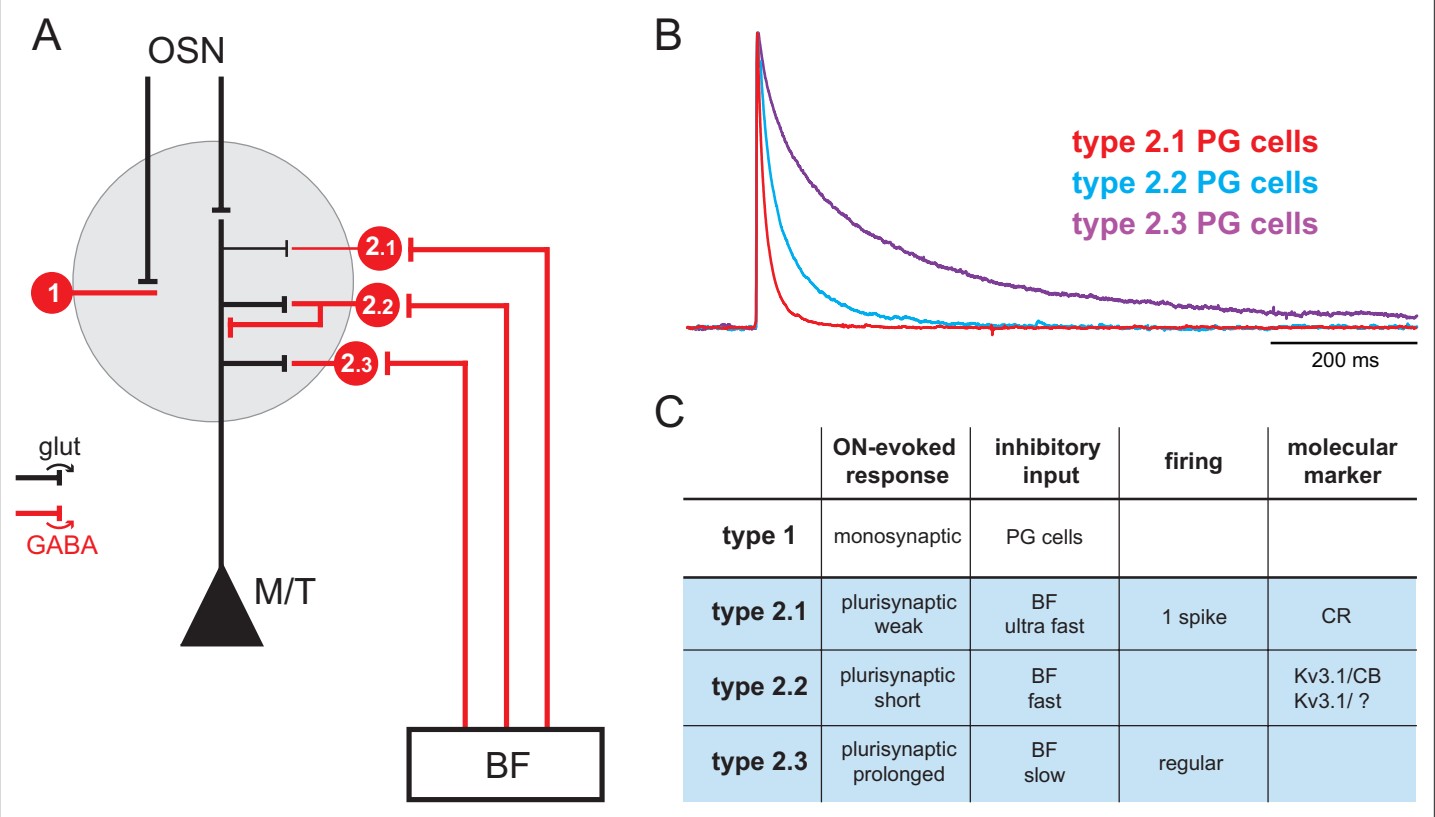

**Figure 1.** Basal forebrain (BF) GABAergic inputs define three subtypes of periglomerular (PG) cells. (**A**) Diagram of the glomerular microcircuit. PG cells are small GABAergic interneurons (red) surrounding each glomerulus. Olfactory sensory neurons (OSNs)-innervated type 1 PG cells do not receive BF GABAergic inputs, whereas type 2 PG cells do. Three subtypes of type 2 PG cells have been identified in *Sanz Diez et al., 2019* based on their synaptic, molecular, and intrinsic properties. See 'Introduction' for details. (**B**) Representative optogenetically evoked BF GABAergic inputs in the different subtypes of type 2 PG cells. Amplitudes are normalized for comparison. Data from *Sanz Diez et al., 2019*. (**C**) Summary table of the properties that distinguish different PG cell subtypes. Cases are left empty when the properties have not been determined, are diverse, or uncertain.

marker, remarkably slow BF IPSCs, and distinctive prolonged ON-evoked plurisynaptic excitatory responses (*Sanz Diez et al., 2019*). Their output connections have not been determined.

I used optogenetic stimulations in OB slices to induce synaptic release of ACh and/or GABA from BF fibers and patch-clamp recording to examine PG cells responses. The results demonstrate that PG cell subtypes are differentially controlled by BF afferents and reveal that the previously overlooked type 2.3 subtype is a central player in mediating BF muscarinic modulation of glomerular inhibition.

## Results

### Synaptic release of ACh activates a subset of PG cells

ChR2 fused with eYFP was first targeted to BF cholinergic neurons by injecting a viral construct into the HDB/MCPO of *Chat*^Cre mice (*Figure 2A*). Choline acetyltransferase (ChAT) immunodetection on brain sections from these mice (hereafter called ChAT mice) confirmed the expression of ChR2 in neurons expressing endogenous ChAT (*Figure 2B*). ChR2 expression, as indicated by eYFP labeling in the HDB/MCPO, was almost exclusively found in cholinergic neurons (89% of the cells positive for eYFP were also ChAT+, 440 double+ cells from a total of 495 eYFP+ cells in coronal slices from four mice) with an infection rate of 34% (440 double+ cells from a total of 1305 ChAT+ cells). Consistent with previous studies (*Hamamoto et al., 2017*; *Kasa et al., 1995*; *Rothermel et al., 2014*; *Smith et al., 2015*), eYFP-labeled cholinergic axons arising from BF cholinergic neurons densely innervated the OB and were abundant in the glomerular layer (*Figure 2C*, *Figure 2—figure supplement 1*).

Next, I used loose cell-attached (LCA) recording in acute OB slices from these mice to monitor spiking activity in randomly chosen PG cells while BF cholinergic axons were periodically stimulated

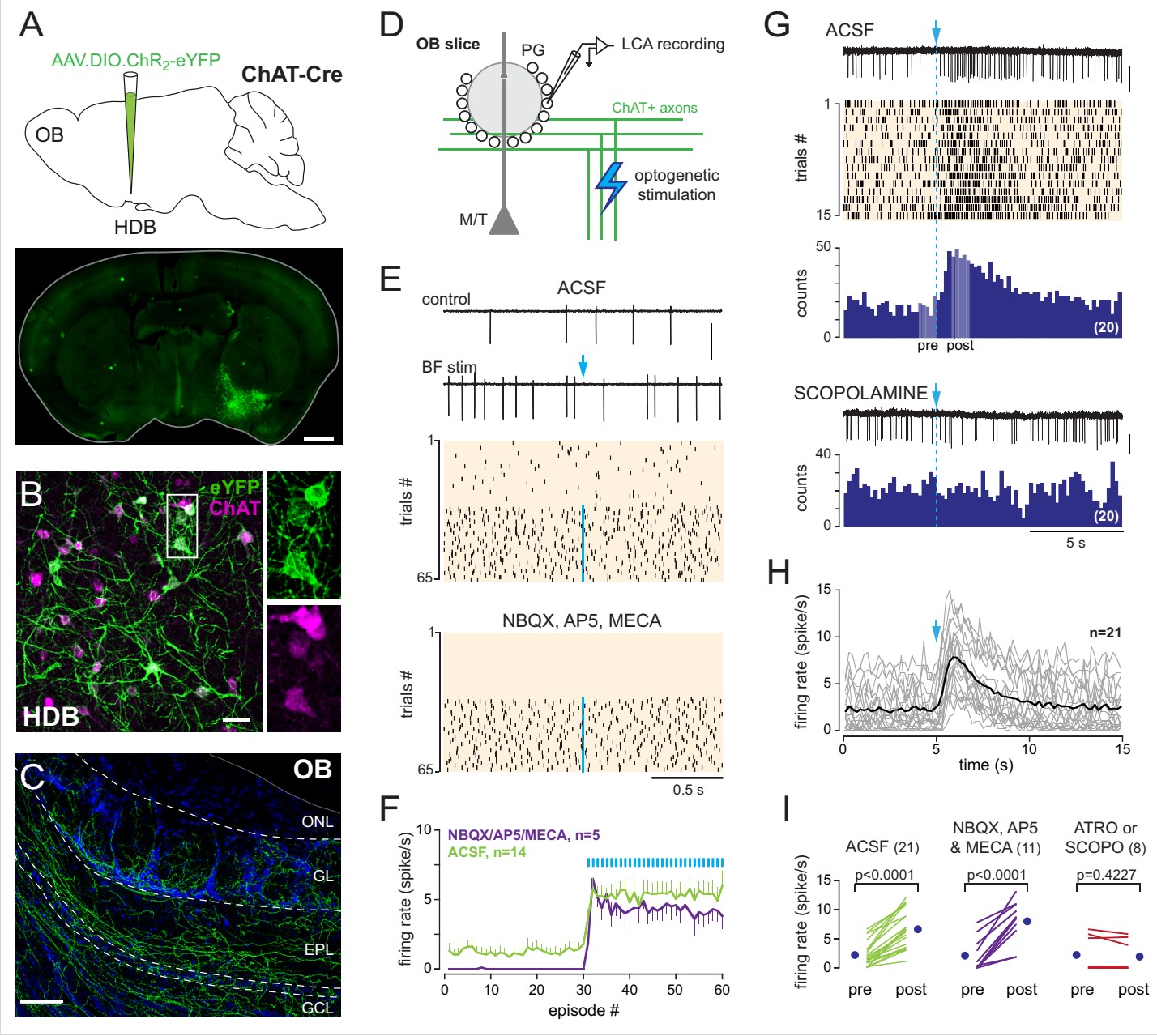

**Figure 2.** Basal forebrain (BF) cholinergic neurons activate a subset of periglomerular (PG) cells. (**A**) Schematic of the virus injection in *Chat^Cre* mice (top) and coronal section (bottom), at about bregma –0.1 mm, 20 days after injection. (**B**) ChR2-eYFP expression (green) in choline acetyltransferase (ChAT)-expressing cholinergic neurons (magenta) in the BF. Scale bar 50 µm. Right panels: zoom on the boxed region. (**C**) ChR2-eYFP-expressing axons in the olfactory bulb (OB). Scale bar 100 µm. DAPI staining (blue) delimits layers (ONL: olfactory nerve layer; GL: glomerular layer; EPL: external plexiform layer; GCL: granule cell layer). Higher-resolution image in *Figure 2—figure supplement 1*. (**D**) Experimental design for recording BF-evoked responses in OB slices. (**E**) Top: two representative 2-s-long loose cell-attached (LCA) recording episodes (scale bar 200 pA) and raster plot of spiking activity in control condition and when BF axons were photostimulated every 2 s (blue arrow, 1 ms flash, 490 nm). Photostimulation started at episode 31 (one flash/episode, blue line). Bottom: raster plot for the same cell, same experiment in the presence of blockers. (**F**) Average firing rate per episode (2 s each) in artificial cerebrospinal fluid (ACSF) (green) or in the presence of blockers (violet). Low-frequency photostimulation (0.5 Hz) started at episode 31. (**G**) Representative recording (scale bar 50 pA), raster plot, and peri stimulus time histograms (PSTH) (20 consecutive trials, bin 200 ms) of an excitatory response evoked in ACSF by a single photostimulation of the BF cholinergic axons at blue arrow and dotted line. Pale areas within the PSTH indicate the two periods that were compared in (**I**). The nonselective muscarinic ACh receptor (mAChR) antagonist scopolamine (10 µM) blocked the evoked excitation (bottom, scale bar 50 pA). (**H**) Average firing rate per bin and per 15-s-long episode for 21 cells recorded in ACSF. Each gray line corresponds to a cell; the black line indicates the ensemble average. Photostimulation at blue arrow. (**I**) Firing rate before (pre) and after (post) photostimulation of BF axons in ACSF (green) in the presence of 6-nitro-7-sulfamoylbenzo[f]quinoxaline-2,3-dione (NBQX), D-2-amino-5-phosphonopentanoic acid (D-AP5), and

*Figure 2 continued on next page*

Figure 2 continued

mecamylamine (violet) or in the presence of the mAChR antagonist atropine (n = 4) or scopolamine (n = 4) (red). Each line indicates a cell; blue circles are the means. Paired *t*-test or Wilcoxon signed-rank-sum test (for atro/scopo).

The online version of this article includes the following figure supplement(s) for figure 2:

**Figure supplement 1.** Higher-resolution image of *Figure 2C* showing the distribution in the olfactory bulb (OB) of choline acetyltransferase (ChAT) mice of eYFP-expressing axons from basal forebrain (BF) cholinergic neurons.

every 2 s (0.5 Hz) using a single brief (1 ms) flash of blue light (*Figure 2D*). This photostimulation induced a significant increase in baseline spike frequency in 7% of the PG cells tested (n = 23/350). No other kind of response was observed. It is noteworthy, however, that more than half of the cells tested were silent or fired only rarely. A cholinergic input may simply be unnoticeable in these cells using LCA spike recording. Responsive PG cells had a low baseline spiking activity in control condition (range 0–6 Hz, mean 1.2 ± 1.6 Hz, n = 14) and switched to a higher-frequency firing mode (mean frequency 5.3 ± 2.7 Hz, n = 14) as soon as low-frequency photostimulations started. This sustained spiking regime was maintained throughout the trials with a stimulation (>1 min) (*Figure 2E and F*). Photo-evoked responses persisted in the presence of 6-nitro-7-sulfamoylbenzo[f]quinoxaline-2,3-dione (NBQX) (10 μM), D-2-amino-5-phosphonopentanoic acid (D-AP5) (50 μM), and mecamylamine (50 μM) (n = 5, *Figure 2E and F*), which inhibit AMPA, NMDA, and nicotinic ACh receptors, respectively. This cocktail of antagonist blocks a possible direct nicotinic excitation of PG cells (*Castillo et al., 1999*), as well as a putative indirect glutamatergic excitation following the nicotinic activation of mitral and tufted cells (*Liu et al., 2015*).

I then used a longer (15–20 s) interval between each flash to examine the time course of the excitatory response. A single flash evoked a reliable and long-lasting increase in firing rate (*Figure 2G*). Spike rate increased on average threefold compared to baseline activity (from 2.2 ± 2.1 Hz before the flash to 6.6 ± 3.3 Hz after the flash, n = 21, *Figure 2I*), peaked about 1 s after the flash, and returned to baseline frequency after about 5–10 s (*Figure 2H*). Evoked excitation persisted in the presence of NBQX, D-AP5 and mecamylamine (spike frequency increased from 2.1 ± 2.8 Hz before the flash to 8.0 ± 3.5 Hz after the flash, n = 11, *Figure 2I*) and was totally blocked by atropine (10 μM, n = 4) or scopolamine (10 μM, n = 4), two nonselective antagonists of metabotropic muscarinic ACh receptors (mAChRs) (*Figure 2G and I*). Thus, transient temporally and spatially precise synaptic release of ACh from BF cholinergic axons strongly excites a subset of PG cells via the activation of mAChRs. The principal focus of this study is on this previously unknown cholinergic pathway.

## PG cells excited by a muscarinic input also receive an inhibitory GABAergic input from separate BF neurons

Nearly all cholinergic neurons in the BF express the molecular machinery to both synthetize and package GABA into synaptic vesicles (*Saunders et al., 2015*). Synaptic co-transmission of GABA from cholinergic axons would be expected to block or reduce spiking if GABA is inhibitory. If GABA is excitatory, co-release of GABA would be expected to trigger spikes even in the presence of AChR antagonists. None of these possible GABAergic responses was seen in ChAT mice, suggesting that GABA is not co-transmitted together with ACh. However, PG cells may still receive a GABAergic input from separate BF neurons. To test this possibility, a virus encoding ChR2-eYFP was injected into the HDB/MCPO of *dlx5/6*Cre mice, a transgenic line that expresses the Cre recombinase in neurons originating from the embryonic ganglionic eminence during development, which includes GABAergic as well as cholinergic neurons in the forebrain (*Monory et al., 2006*; *Figure 3A*). Accordingly, we showed in our previous study that recombination induces the expression of ChR2-eYFP in several populations of GABAergic neurons as well as in ChAT-expressing cholinergic neurons in the BF (*Sanz Diez et al., 2019*).

Axonal projections of ChR2-eYFP-expressing neurons densely innervated the OB in dlx5/6 mice, especially in the glomerular layer and the granule cell layer (*Figure 3B*, *Figure 3—figure supplement 1*). This projection pattern is consistent with earlier reports in GAD2-cre mice (*Böhm et al., 2020*; *Villar et al., 2021*) or vGAT-cre mice (*Hanson et al., 2020*), two other lines often used to label BF GABAergic neurons. In OB slices from dlx5/6 mice, 0.5 Hz photostimulations affected post-stimulus spiking activity in 30% of the cells tested (116/383). Responses were heterogeneous, but three clearly

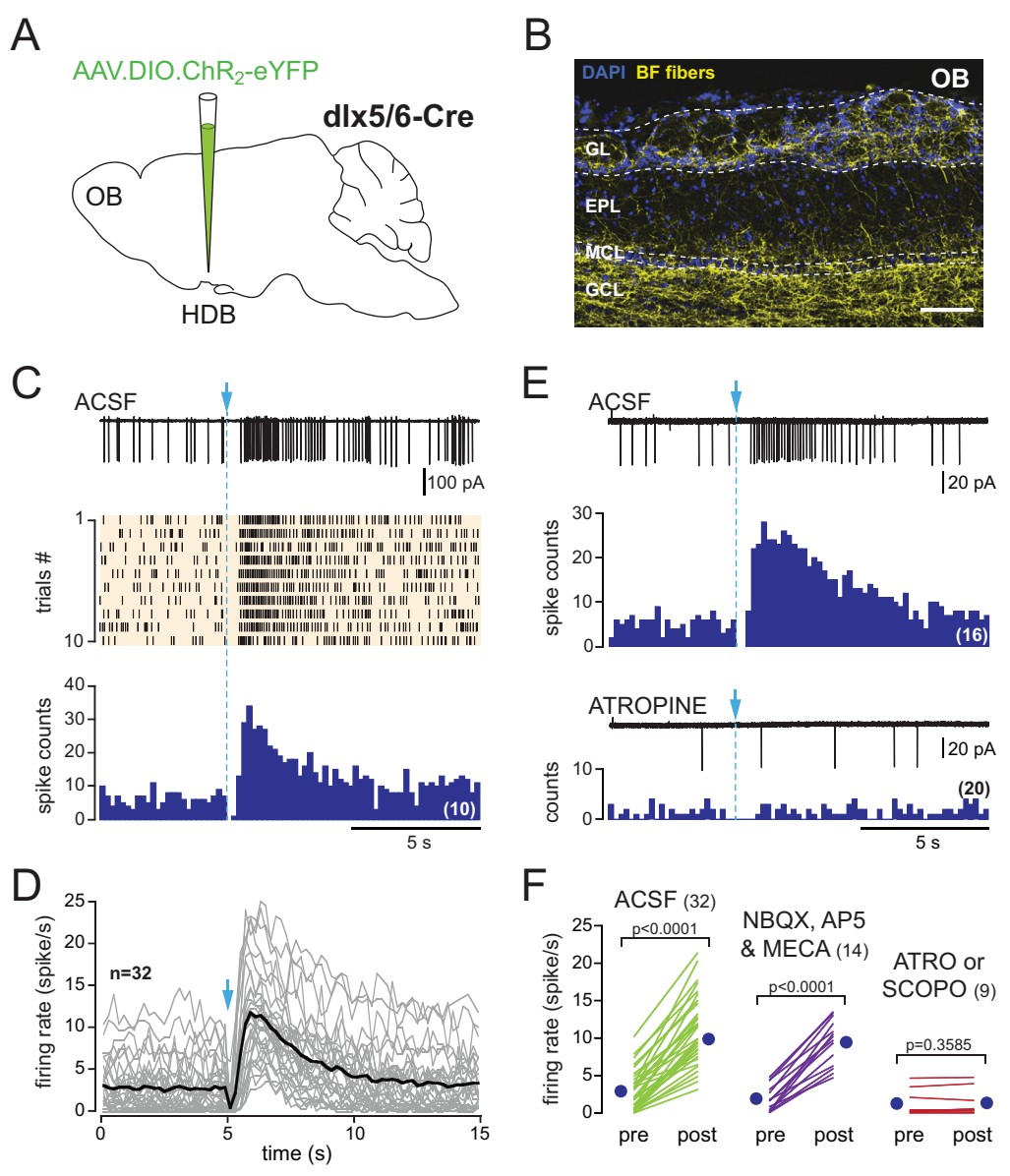

**Figure 3.** In dlx5/6 mice, a single photostimulation of basal forebrain (BF) axons evokes a biphasic inhibition-excitation response in periglomerular (PG) cells with a muscarinic excitation. (**A**) Schematic of the virus injection in the horizontal limb of the diagonal band of Broca/magnocellular preoptic nucleus (HDB/MCPO) of *dlx5/6*[Cre] mice. (**B**) ChR2-eYFP expression in BF axons (yellow) in a sagittal section of the olfactory bulb (OB). DAPI staining (blue) delimits the different layers (GL: glomerular layer; EPL: external plexiform layer; MCL: mitral cell layer; GCL: granule cell layer). Scale bar 100 μm. Higher-resolution image in *Figure 3—figure supplement 1*. (**C**) Representative spiking response, raster plot, and cumulative peri stimulus time histogram (PSTH) (10 consecutive sweeps, 200 ms/bin) of a typical biphasic inhibition-excitation response evoked by a single photostimulation of BF fibers and recorded over 15 s in a PG cell from a dlx5/6 mouse. (**D**) Average spiking frequency per bin (200 ms) and per episode. Each gray line corresponds to a cell. The black line is the ensemble average. Photostimulation at blue arrow. Only one cell in the dataset responded with a long-lasting excitation that was not preceded by an inhibitory component (*Figure 3—figure supplement 3*). (**E**) The nonselective muscarinic ACh receptor (mAChR) antagonist atropine (10 μM) blocked BF-evoked excitation. (**F**) Firing rate before (pre) and after (post) photostimulation of BF axons in artificial cerebrospinal fluid (ACSF) (green) in the presence of 6-nitro-7-sulfamoylbenzo[f]quinoxaline-2,3-dione (NBQX), D-2-amino-5-phosphonopentanoic acid (D-AP5), and mecamylamine (violet) or in the presence of the muscarinic receptor antagonist atropine (n = 7) or scopolamine (n = 2) (red). Each line indicates a cell. Blue circles indicate means. Paired *t*-test.

*Figure 3 continued on next page*

*Figure 3 continued*

The online version of this article includes the following figure supplement(s) for figure 3:

**Figure supplement 1.** Higher-resolution image of *Figure 3B* showing the distribution in the olfactory bulb (OB) of a dlx5/6 mouse of eYFP-expressing axons from basal forebrain (BF) cholinergic and GABAergic neurons.

**Figure supplement 2.** Basal forebrain (BF) inputs have various impacts on periglomerular (PG) cells activity in dlx5/6 mice.

**Figure supplement 3.** the unique example, in a dlx5/6 mouse, of a cell responding with a long-lasting excitation that was not accompanied by a transient inhibitory phase immediately after the flash.

distinct types of responses were evoked (*Figure 3—figure supplement 2*): 12% of the cells tested (45/383) responded with a transient inhibition of spiking immediately after the stimulation, as expected for a classical inhibitory input (see also Figure 8). 7% of the cells responded with a brief excitation (29/383 cells), that is, the photostimulation induced a single spike, sometimes two (see also Figure 9). Finally, 11% of the cells (42/383 cells) responded with a transient inhibition of spiking after each flash and with a robust and long-lasting increase in baseline firing rate as soon as low-frequency photo-stimulations started. This last group of cells with dual responses likely correspond to those receiving a muscarinic excitation. Consistent with this, a single flash evoked a robust and long-lasting increase in firing rate in these cells (*Figure 3C*), similar to in ChAT mice. This excitation persisted in the presence of NBQX, D-AP5, and mecamylamine and was blocked by atropine (n = 7) or scopolamine (n = 2) (*Figure 3C–F*). However, contrasting with the exclusive excitatory muscarinic responses in ChAT mice, spiking was also transiently blocked after the photostimulation. PG cells with these dual responses had a low and irregular firing activity in control condition (range 0–4 Hz, mean 1.95 ± 1.6 Hz, n = 13) and switched to a sustained higher-frequency firing mode upon the 0.5 Hz low-frequency photostim-ulations in artificial cerebrospinal fluid (ACSF) (mean 8.6 ± 3 Hz, n = 13, *Figure 4A–C*). This sustained spiking regime was also induced when low-frequency stimulations were done in the presence of NBQX, D-AP5, and mecamylamine (n = 10, *Figure 4D–F*). When delivered every 2 s, each flash was followed by a period of spiking inhibition (*Figure 4A–E*). Photo-evoked spiking inhibition lasted 391 ± 142 ms (n = 21), persisted in the presence of NBQX, D-AP5, and mecamylamine (n = 10, *Figure 4D–F*), and was blocked by the $GABA_A$ receptor antagonist gabazine (5 µM, *Figure 4G and H*). Gabazine did not prevent the ACh-mediated increase of basal firing evoked by the 0.5 Hz photostimulations (n = 8, *Figure 4G–I*). Only one cell with the muscarinic excitatory response lacked the GABAergic compo-nent in dlx5/6 mice (*Figure 3—figure supplement 3*). Although rare, this single case confirms that the cholinergic neurons stimulated in dlx5/6 mice do not co-transmit ACh and GABA. Altogether, the results are consistent with previous investigations demonstrating that although BF cholinergic neurons express molecular markers of GABAergic neurons, only a fraction of them co-releases GABA and ACh (*Saunders et al., 2015*). They further indicate that a subset of PG cells receives both ACh and GABA inputs from separate BF axons. GABA release inhibits spiking through $GABA_A$ receptors, whereas ACh release activates mAChRs and produces a long-lasting excitation.

## Endogenous ACh release elicits a slow muscarinic excitatory postsynaptic current (EPSC in type 2.3 PG cells)

In our previous report, we did not detect any BF-evoked muscarinic EPSC in whole-cell (WC) recordings from various PG cell subtypes in dlx5/6 mice (*Sanz Diez et al., 2019*). To understand the mechanism underlying photo-evoked muscarinic excitation, I re-examined this question using a slightly different K-gluconate-based internal solution supplemented with phosphocreatine to improve ATP supply. PG cells receiving a muscarinic input were first identified using LCA spike recording and then WC patched with a pipette filled with the new internal solution (n = 5 cells in ChAT mice and n = 12 in dlx5/6 mice). In dlx5/6 mice, a single photostimulation evoked a biphasic response initiated by an outward IPSC followed by a slow-rising inward current in 7/12 cells (*Figure 5A*). The slow EPSC was undetectable in five other cells. In ChAT mice, BF axons photostimulation induced a slow inward EPSC in 5/5 cells. Consistent with the LCA experiments, the slow inward current was not preceded by an outward IPSC, further confirming that cholinergic axons do not release GABA on these PG cells (*Figure 5D*). In both transgenic lines, photo-evoked muscarinic EPSCs recorded at negative holding potentials (from –30 to –60 mV) were small and often at the limit of detection (max 10 pA, mean 5 ± 3 pA, data obtained in

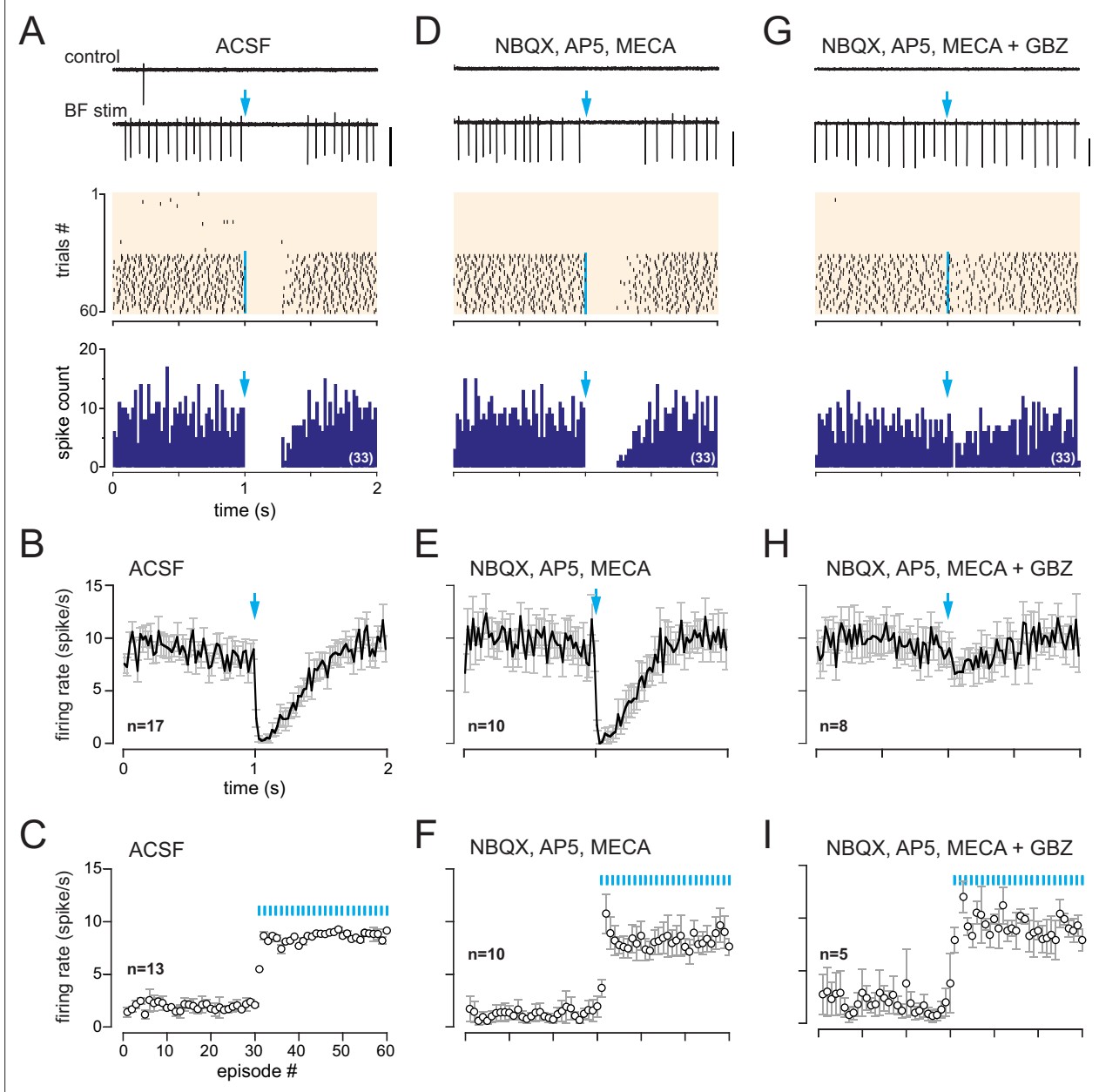

**Figure 4.** In dlx5/6 mice, a monosynaptic GABAergic input inhibits spiking in cells with a muscarinic response. (**A**) Two representative traces (scale bar 100 pA) showing spiking activity in control condition and while basal forebrain (BF) axons were photostimulated with a single flash per episode every 2 s (blue arrow). Middle: corresponding raster plot. Photostimulations started at episode 31 (blue line). Bottom: cumulative peri stimulus time histogram (PSTH) (bin size 20 ms) in the same cell for the trials with a photostimulation (blue arrow). (**B**) Average firing rate per bin (20 ms) and per episode for 17 cells (± SEM, gray bars) while BF axons were photostimulated every 2 s with a single flash (at blue arrow). (**C**) Average firing frequency per episode (2 s each) for 13 cells recorded in artificial cerebrospinal fluid (ACSF) in control condition (no light, episodes 1–30) and during photostimulation of the BF afferents once per episode (31–60). (**D–F**) Same as in (**A–C**) in the presence of 6-nitro-7-sulfamoylbenzo[f]quinoxaline-2,3-dione (NBQX) (10 µM), D-2-amino-5-phosphonopentanoic acid (D-AP5) (50 µM),, and mecamylamine (MECA, 50 µM). Traces, raster plot, and PSTH in (**D**) are from the same cell as in (**A**). (**G–I**) Same as in (**A–C**) when gabazine (GBZ, 5 µM) was added to the cocktail of blockers. Traces, raster plot, and PSTH in (**G**) are from the same cell as in (**A**) and (**D**).

ChAT and dlx5/6 mice pooled together). In current-clamp, at membrane potential close to the resting potential, photo-evoked excitatory postsynaptic potentials (EPSPs did not exceed 5 mV (mean 3.1 ± 1.1 mV, two cells in ChAT mice and three cells in dlx5/6 mice)) (*Figure 5A and D*). Unfortunately, muscarinic EPSC/EPSP ran down in few minutes, precluding any further characterization.

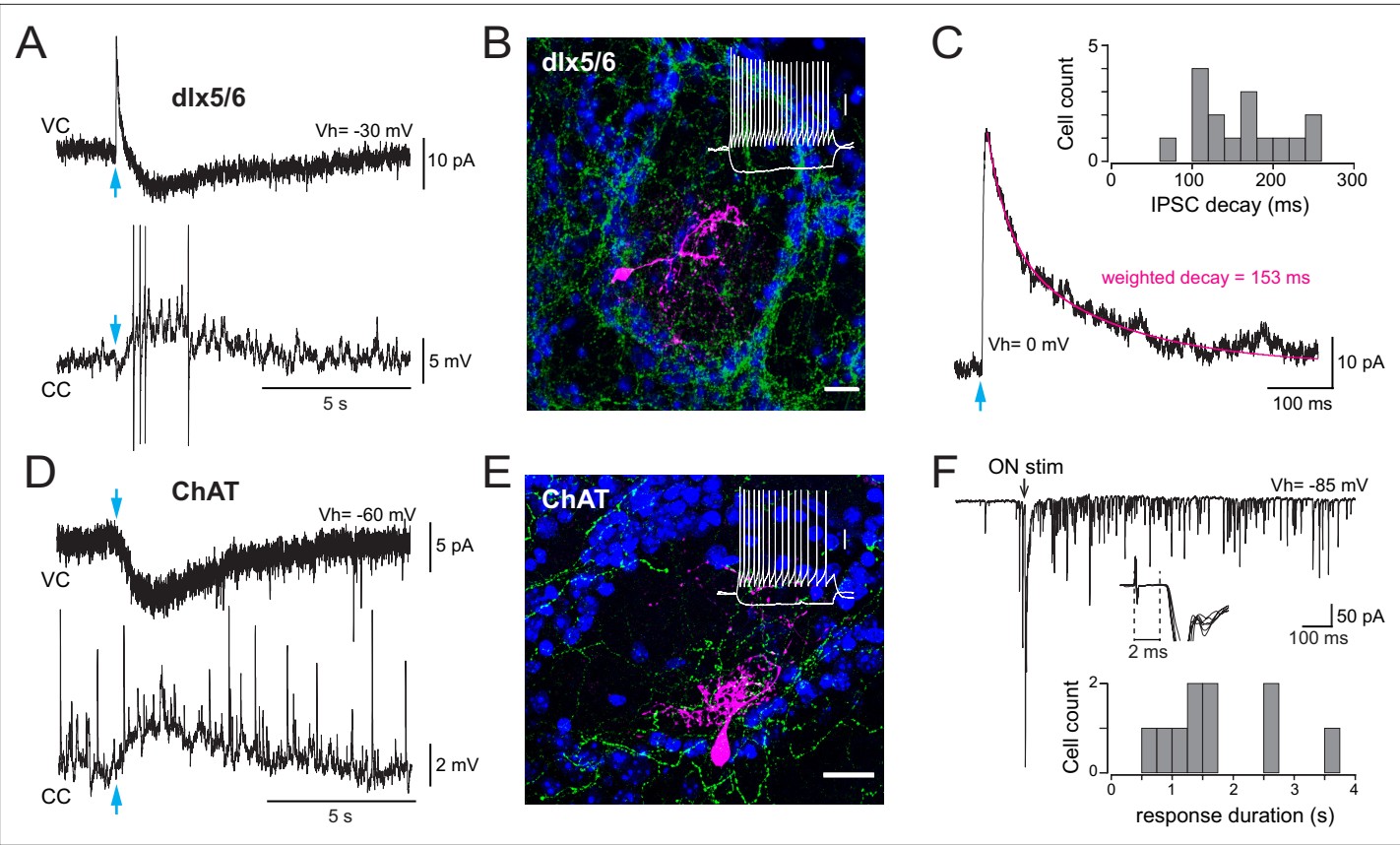

**Figure 5.** Basal forebrain (BF) cholinergic inputs produce a slow muscarinic EPSC in type 2.3 periglomerular (PG) cells. (**A**) Biphasic GABAergic-muscarinic voltage-clamp (VC) and current-clamp (CC) whole-cell responses to a single photostimulation (blue arrow) of BF axons in a cell from a dlx5/6 mouse. Responses recorded in the presence of 6-nitro-7-sulfamoylbenzo[f]quinoxaline-2,3-dione (NBQX), D-2-amino-5-phosphonopentanoic acid (D-AP5), and mecamylamine. The current is an average of eight consecutive sweeps, the voltage response is a single trace eliciting four spikes (truncated for display). (**B**) Morphology of a biocytin-filled PG cell in which photostimulation of BF axons (eYFP-positive, green) produced a dual GABA-ACh response in a dlx5/6 mouse. DAPI staining (blue) shows the outline of the glomerulus. Scale bar 20 µm. Inset: voltage responses of this cell to depolarizing and hyperpolarizing current steps (20 pA, 500 ms). Scale bar 20 mV. (**C**) Photo-evoked GABAergic IPSC recorded in the same PG cell as in (**A**) at a holding potential of 0 mV. Average of 10 sweeps. The decay was best fitted with two exponentials (magenta) with a weighted decay time constant of 153 ms. Inset: distribution histogram of the decay time constants of photo-evoked GABAergic IPSCs in PG cells with a mixed GABA/ACh response in dlx5/6 mice. (**D**) Photo -evoked muscarinic EPSC (VC) and EPSP (CC) recorded in artificial cerebrospinal fluid (ACSF) in a PG cell from a choline acetyltransferase (ChAT) mouse. Average of six consecutive sweeps for the EPSC, single trace for the EPSP. (**E**) Morphology of a biocytin-filled cell that responded to the photostimulation of BF cholinergic axons with a muscarinic excitation in a ChAT mouse. Scale bar 20 µm. Blue: DAPI; green: eYFP-positive BF axons. Inset: membrane voltage responses of this cell to the injection of current steps (–20/+ 35 pA, 500 ms). Scale bar 20 mV. (**F**) Long-lasting barrage of EPSCs evoked by an electrical stimulation of the olfactory nerves (black arrow, 0.1 ms/100 µA) in the cell shown in (**E**). Inset: zoom on the earliest phase of the response (six consecutive traces, truncated for display). Responses had onset latencies > 2 ms. The distribution histogram shows the duration of the olfactory nerve (ON)-evoked response elicited in 10 cells receiving a muscarinic excitation (seven cells from dlx5/6 mice and three cells from ChAT mice).

Intrinsic, synaptic, and morphological properties of the ACh-responsive PG cells were also examined. In addition to the 17 WC-recorded cells from ChAT and dlx5/6 mice, the dataset included WC recordings from previous experiments in dlx5/6 mice in which a photo-evoked muscarinic excitation was noticed in the cell-attached configuration but not detected in the WC mode (n = 9). The morphology of each cell was assessed at the end of the recording by visual inspection of the dye present in the internal solution. Moreover, six cells filled with biocytin were successfully recovered for post-hoc morphological reconstruction. All had the typical morphology of PG cells. Their soma was small, ovoid, or round, with no apparent axon. Thin dendrites projected within a single glomerulus (*Figure 5B and E*). On average, their electrical membrane resistance was 1061 ± 524 MΩ. They all fired regularly at up to 100 Hz with overshooting action potentials in response to depolarizing current steps (*Figure 5B and E*). Another striking hallmark of these PG cells was their prolonged response to

a single electrical stimulation of the ONs. ON-evoked response consisted of a barrage of fast EPSCs lasting several hundreds of ms (average 1760 ± 927 ms, n = 10) (*Figure 5F*). Evoked responses had an onset latency >2 ms (average 3.14 ± 0.66 ms) as typically seen in pluri-synaptic ON-evoked responses of type 2 PG cells (*Sanz Diez et al., 2019*; *Najac et al., 2015*). Finally, in dlx5/6 mice, photo-evoked BF IPSCs (amplitude range 10–186 pA, mean 62 ± 40 pA) were particularly slow (average decay time constant 162 ± 55 ms, range 69–258 ms, n = 16) (*Figure 5C*). Together, these intrinsic and synaptic properties unambiguously identify type 2.3 PG cells as the principal and perhaps unique synaptic target of BF cholinergic axons among PG cells.

To confirm that type 2.3 PG cells are the only PG cell target of BF cholinergic neurons, I examined WC responses evoked by a single flash in other types of PG cells in ChAT mice (n = 54). Photo-evoked responses were recorded in voltage-clamp at 0 mV and at negative holding potential in order to detect possible monosynaptic GABAergic, nicotinic, or muscarinic responses. Membrane properties and ON-evoked responses were examined when possible to identify each cell type. Photostimulation of the BF fibers did not evoke any response in type 1 PG cells (n = 4), type 2.1 PG cells (n = 21), and type 2.2 PG cells (n = 17). 12 additional PG cells that could not be firmly classified did not respond either. Only one PG cell, which could not be unambiguously classified, responded with a fast IPSC, a rare event that possibly reflected unspecific expression of ChR2 in BF GABAergic neurons. Together, these results indicate that BF cholinergic axons innervating the glomerular layer of the OB have a unique and selective synaptic target among PG cells and do not release GABA.

## M1 mAChR mediates the muscarinic excitation

Next, I determined what muscarinic receptor mediates the cholinergic response in type 2.3 PG cells and what downstream mechanism produces the excitation. There are five types of mAChRs, and three of those (M1, M3, and M5) are coupled to excitatory $G_{q/11}$ proteins that activate PLCβ, causing hydrolysis of PIP2 into DAG and IP3. Activation of these receptors most often depolarizes neurons. M1 receptors are widely expressed in the OB (*Le Jeune et al., 1995*) and their blockage impairs olfactory-evoked fear learning (*Ross et al., 2019*), making them a likely candidate. To test this hypothesis, I examined the effects of pirenzepine (1–2 µM), a selective antagonist of M1 receptors, on BF-induced muscarinic excitation in both ChAT (n = 3 cells) and dlx5/6 mice (n = 7). Experiments were done using LCA recording in the presence of glutamate and nicotinic receptor antagonists. Pirenzepine fully blocked the excitation evoked by light stimulation of the BF fibers in six cells and reduced its strength in four cells (*Figure 6A and B*). On average, post-stimulation spike frequency was reduced four-fold by pirenzepine (control: 10 ± 3.4 Hz; pirenzepine: 2.6 ± 3.3 Hz, p<0.0001, data from ChAT and dlx5/6 mice pooled together, *Figure 6B*). These results support the idea that M1 receptors mediate the muscarinic excitation of type 2.3 PG cells.

Could M1 receptors serve as a molecular marker for type 2.3 PG cells? To address this question, I examined the distribution of an mRNA transcript of Chrm1, the gene encoding M1 mAChRs, using RNAscope fluorescence in situ hybridization (FISH) in OB sections. Hybridization signals were found around DAPI-stained nuclei in all layers of the OB (*Figure 6—figure supplement 1A and B*). This widespread distribution is consistent with an earlier autoradiography binding study (*Le Jeune et al., 1995*) and with more recent ISH data in the Allen Brain Atlas (https://mouse.brain-map.org/experiment/show/73907497; *Lein et al., 2007*) or immunohistochemical data in the Human Protein Atlas (https://www.proteinatlas.org/ENSG00000168539-CHRM1/brain; *Sjöstedt et al., 2020*). The signal was particularly strong in granule cells (*Figure 6—figure supplement 1B–D*), consistent with previous functional data demonstrating that M1 mAChRs potentiate the excitability of granule cells (*Pressler et al., 2007*). However, all the granule cells recorded in this study in ChAT mice in the LCA mode were silent and remained silent during the low-frequency photostimulations of the cholinergic fibers (n = 17, not shown). M1 could still be a useful molecular marker of type 2.3 PG cells if it is strongly and consistently expressed in these cells and have minimal expression in other cell types in the glomerular layer. However, most cells around glomeruli expressed the mRNA transcript of Chrm1. Cells with no signal were more abundant in the glomerular layer than in the granule cell layer but still represented a small fraction of the cells (*Figure 6—figure supplement 1C and D*). Moreover, ISH combined with immunodetection of CR or tyrosine hydroxylase (TH), two commonly used markers of nonoverlapping juxtaglomerular cell types, revealed that the Chrm1 transcript was expressed in 85% of the CR-positive (n = 372 cells counted) and 94% of the TH-positive cells (n = 181 cells) (*Figure 6—figure*

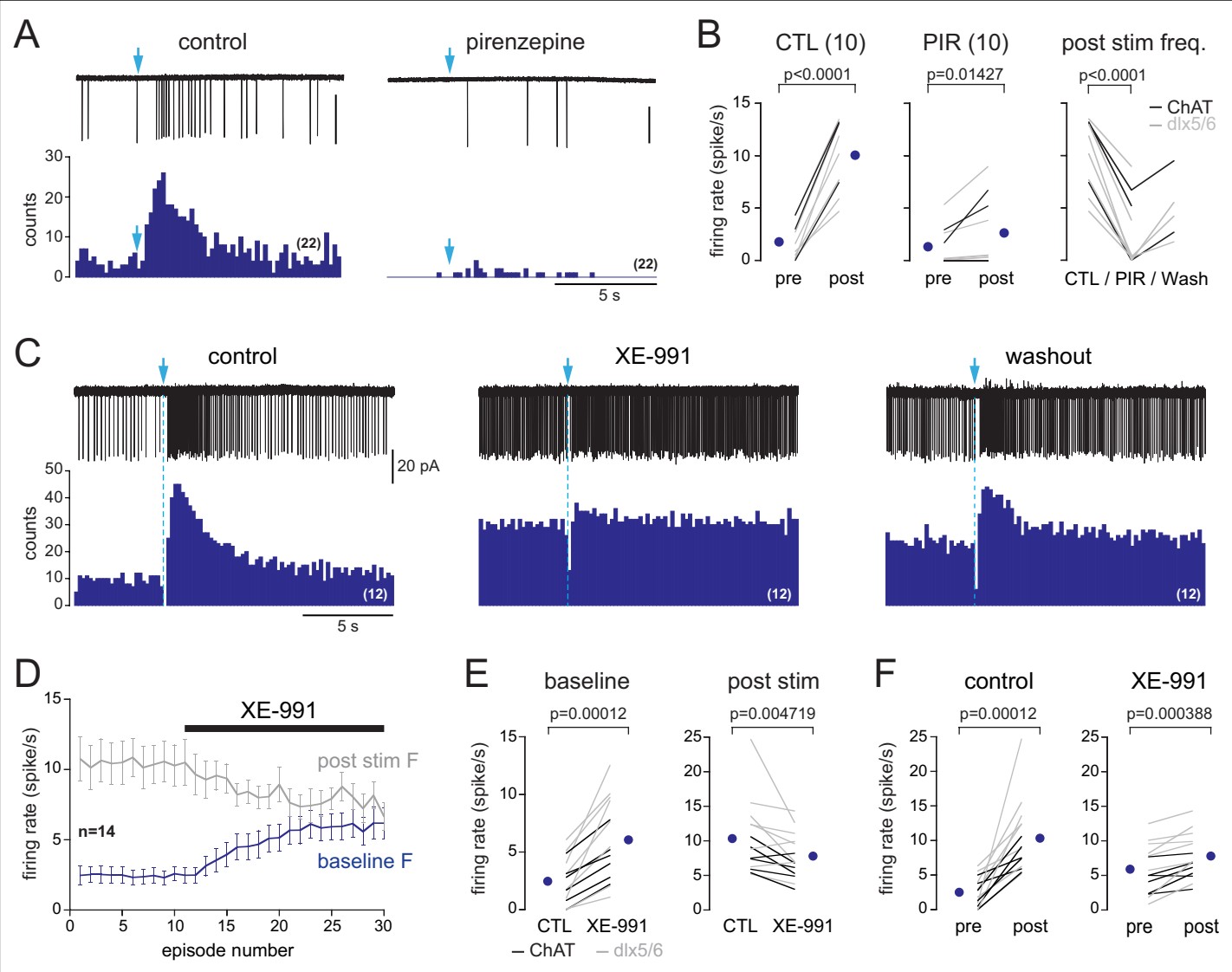

**Figure 6.** M1 muscarinic ACh receptors (mAChRs) mediate basal forebrain (BF)-evoked excitation by closing M channels. (**A**) Photo-evoked loose cell-attached (LCA) responses and cumulative peri stimulus time histograms (PSTHs) (over 22 consecutive sweeps, 200 ms/bin) recorded in a periglomerular (PG) cell from a dlx5/6 mouse in control condition (left) and in the presence of the M1 mAChR antagonist pirenzepine (2 μM). Scale bar for traces 50 pA. (**B**) Summary graphs. Firing rate before (pre) and after (post) photostimulation of BF fibers in control condition (left, paired *t*-test) and in the presence of pirenzepine (PIR, 1 or 2 μM, paired Wilcoxon signed-rank-sum test). Right graph shows that pirenzepine decreased BF-evoked excitation in every cell tested (paired *t*-test). Partial washout was obtained in five cells. Cells were recorded in choline acetyltransferase (ChAT) mice (n = 3, black lines) and dlx5/6 mice (n = 7, gray lines). (**C**) Photo-evoked LCA responses and cumulative PSTHs recorded in a PG cell from a dlx5/6 mouse showing the effects of the M-channel blocker XE-991 (10 μM) on spiking frequency. BF fibers were photostimulated with a single flash (blue arrow and dotted line). (**D**) XE-991 increased baseline spiking rate (blue line, measured during a 15 s time period preceding the flash) and decreased post-stimulus spike frequency (gray line). Average from 14 cells (eight in dlx5/6 mice, six in ChAT mice). Each episode was 30 s long. (**E**) Summary graph showing the two effects of XE-991 on each cell. Paired Wilcoxon signed-rank-sum tests. (**F**) Firing rate before (pre) and after (post) photostimulation of BF fibers in control condition (left, paired Wilcoxon signed-rank-sum test) and in the presence of XE-991 (*t*-test). Experiments were all done in the presence of 6-nitro-7-sulfamoylbenzo[f]quinoxaline-2,3-dione (NBQX) (10 μM), D-2-amino-5-phosphonopentanoic acid (D-AP5) (50 μM), and mecamylamine (50 μM). Means are the blue circles.

The online version of this article includes the following figure supplement(s) for figure 6:

**Figure supplement 1.** RNAscope imaging of Chrm1 mRNA expression in the olfactory bulb (OB).

supplement 1E and F). These results thus suggest that multiple cell types in the glomerular layer express M1 receptors.

M1 receptors classically suppress the M current, a slow voltage-activated potassium current mediated by KCNQ2/3 potassium channels and active at resting membrane potential (*Brown and*

*Passmore, 2009*; *Suh and Hille, 2008*). KCNQ2 subunits are strongly expressed in PG cells (*Cooper et al., 2001*). The fast run-down of the BF-evoked muscarinic EPSC in WC recordings as well as its strong dependence on intracellular ATP is also consistent with this downstream mechanism. Cell dialysis is indeed often deleterious for M currents because M-channels opening depends on PIP2 binding, a process that is itself highly dependent on intracellular ATP supply for PIP2 phosphorylation (*Suh and Hille, 2008*; *Zhang et al., 2003*). To investigate whether closure of the M current causes the muscarinic depolarization of type 2.3 PG cells, I applied the selective M-channel antagonist XE-991 (10 µM) in the presence of NBQX, D-AP5, and mecamylamine. XE-991 had two noticeable effects in type 2.3 PG cells recorded in LCA. First, it increased baseline spike frequency in all the cells tested (n = 14, eight in dlx5/6 mice, six in ChAT mice, *Figure 6C–E*), suggesting that M-channels are indeed open at rest in type 2.3 PG cells and hyperpolarize their membrane potential. Second, although XE-991 did not fully block photo-evoked muscarinic excitation in most cells (*Figure 6C and F*), it reduced the strength of BF-induced excitation (*Figure 6D and E*). A persistent photo-evoked excitation is expected if the muscarinic EPSP is partially blocked while the membrane potential is more depolarized compared to control condition. Overall, these data support the hypothesis that activation of M1 mAChRs depolarizes type 2.3 PG cells by blocking a potassium M current.

## Muscarinic excitation of type 2.3 PG cells leads to an increase of inhibitory synaptic inputs in principal neurons

PG cells are classically viewed as a source of inhibitory GABAergic synaptic currents (IPSCs) for mitral and tufted cells, the two OB output channels that project to distinct cortical areas. However, the targets and output properties of type 2.3 PG cells have never been specifically investigated. Thus, to evaluate the impact of the muscarinic excitation of type 2.3 PG cells, I recorded IPSCs in mitral cells and superficial or middle tufted cells (s/mTC) in the presence of NBQX, D-AP5, and mecamylamine (*Figure 7A*). I also examined IPSCs in external tufted cells (eTCs) (*Figure 7A*). Although it is still unclear whether eTCs have axonal projections outside the OB like other tufted cells, they play a major role in processing incoming OSN information by coordinating rhythmic activity within each glomerulus and providing feedforward excitation to various types of neurons, including PG, mitral, and tufted cells (*De Saint Jan et al., 2009*; *Hayar et al., 2004*; *Najac et al., 2011*; *Najac et al., 2015*).

Photostimulation of the cholinergic axons significantly increased IPSCs frequency in eTCs (n = 9) and s/mTCs (n = 10) (*Figure 7*), whereas no response was found in mitral cells (n = 17). However, three of the recorded mitral cells had severed apical dendrites and most of the intact cells (10/14) projected in glomeruli located deep within the slice, where light stimulation may be less efficient. Baseline IPSCs frequency greatly varied across cells and was on average smaller in s/mTC (7.0 ± 5.1 Hz) compared to eTC (15.7 ± 10 Hz, p=0.0285, *t*-test). Yet, IPSCs increased in similar proportion and with comparable time course in both cell types after photostimulation (*Figure 7—figure supplement 1*). Data were thus pooled in *Figure 7D and E*. Thus, low-frequency photostimulation led to a rapid and persistent increase in IPSCs frequency in 8/10 eTC and 10/23 s/mTC. This response was blocked (n = 8) or reduced (n = 3) when the experiment was repeated in the presence of 2 µM pirenzepine (*Figure 7C and E*). Pirenzepine had little effect in 4/15 cells. A single photostimulation also transiently increased IPSCs frequency in eTC (n = 9) and s/mTC (n = 8) (*Figure 7B and D*). Addition of pirenzepine attenuated this response in 10/11 cells. Together, these results suggest that type 2.3 PG cells release GABA onto tufted cells following their excitation by muscarinic activation. This increase in inhibitory synaptic inputs could lead to profound changes in the activity and output of the OB network.

## BF GABAergic inputs inhibit type 2.2 PG cells

Besides type 2.3 PG cells described above, a second group of cells in dlx5/6 mice responded to the optogenetic stimulation of the BF axons with a transient block of spiking immediately after the photostimulation (n = 45/383, 12% of the cells tested) (*Figure 8*). However, unlike type 2.3 PG cells, they did not show any evidence of parallel cholinergic excitation. Baseline firing activity did not rapidly increase when BF fibers were photostimulated at 0.5 Hz and a single flash did not elicit a long-lasting mAChR-evoked increase in firing (*Figure 8—figure supplement 1*). In these cells, basal firing frequency was, on average, higher than in type 2.3 PG cells (mean 7.8 ± 5.6 Hz, n = 21, p=0.00058, Mann–Whitney rank-sum test). However, the nature of this activity (single spike vs. burst of spikes) and its frequency (range 2–21 Hz) varied greatly across cells (*Figure 8—figure supplement 2*). Spontaneous spiking

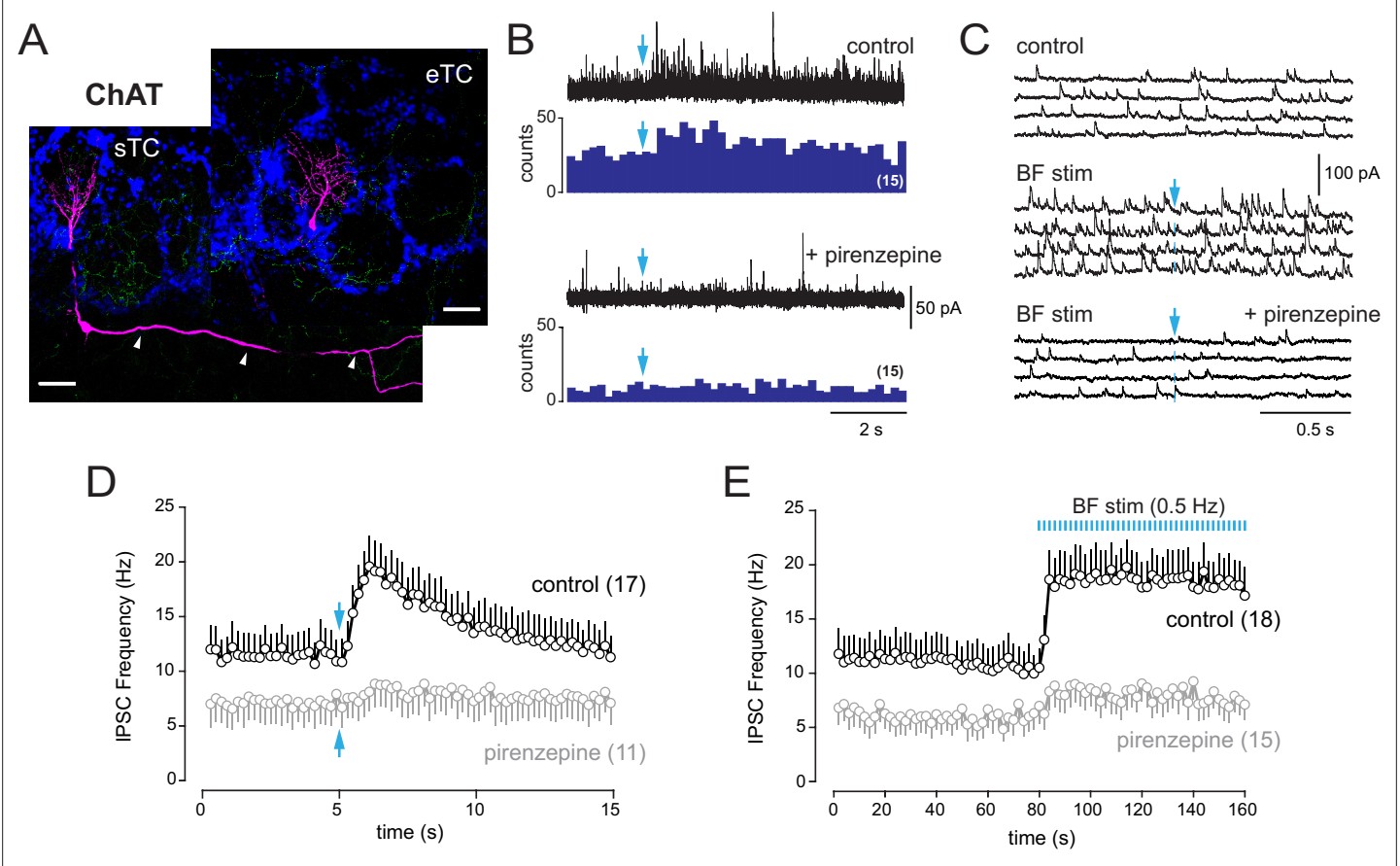

**Figure 7.** Basal forebrain (BF) muscarinic inputs lead to an increase of inhibitory synaptic inputs in tufted cells. (**A**) Morphologies of two biocytin-filled tufted cells recorded in different slices from choline acetyltransferase (ChAT) mice. The localization of the soma and the presence or not of lateral dendrites in the external plexiform layer (arrowheads) distinguish a superficial tufted cell (sTC) (left) and an external tufted cell (eTC) (right). DAPI staining (blue) shows the outline of glomeruli. ChR2-eYFP-expressing cholinergic fibers are visible in green. Scale bars 50 μm. (**B**) Increase of inhibition evoked by a single photostimulation (blue arrow) of the cholinergic axons in an sTC (top). Pirenzepine (2 μM) blocked this response (bottom). Five traces are superimposed in each condition. Peri stimulus time histograms (PSTHs) show the cumulative number of IPSCs/bin (200 ms) across 15 consecutive trials. (**C**) Four consecutive traces of spontaneous IPSCs recorded in an eTC in control condition (top) and during low-frequency photostimulation of the cholinergic BF fibers (one flash every 2 s at blue arrow, middle). Pirenzepine (2 μM) blocked the increase of IPSC frequency evoked by the photostimulations (bottom). (**D**) Average IPSC frequency per 200 ms bin and per episode for 17 tufted cells (eight s/mTC and nine eTC). BF axons were photostimulated once (blue arrow). Pirenzepine was tested on 11/17 cells and reduced photo-evoked increase of IPSCs. (**E**) Average IPSC frequency per episode (2 s) for 18 cells (10 s/mTC and 8 eTC). Photostimulation of the cholinergic fibers at 0.5 Hz rapidly and persistently increased IPSC frequency. Pirenzepine was tested in 15/18 cells. Experiments were done in ChAT mice in the presence of 6-nitro-7-sulfamoylbenzo[f]quinoxaline-2,3-dione (NBQX), D-2-amino-5-phosphonopentanoic acid (D-AP5), and mecamylamine. Individual data points are shown in *Figure 7—figure supplement 1*.

The online version of this article includes the following figure supplement(s) for figure 7:

**Figure supplement 1.** Comparison of basal forebrain (BF)-induced increase of IPSCs in external tufted cells (eTC) and superficial and middle tufted cells (s/mTC).

was also diversely affected by NBQX, D-AP5, and mecamylamine, being totally blocked in some cells (n = 3) and not affected in others (n = 9). In the latter, BF-induced inhibition persisted in the presence of the blockers (n = 9) and was blocked by gabazine (*Figure 8B*, n = 7), suggesting that it was caused by a direct GABAergic BF input. The duration of the BF-induced inhibition also varied across cells. In about half of the cells, the BF input induced a shorter (<200 ms) inhibition than in type 2.3 PG cells, whereas inhibition was longer and similar to in type 2.3 PG cells in others (*Figure 8C*). However, it is noteworthy that cells with a prolonged inhibition often fired at low rate or fired irregularly with bursts of spikes (*Figure 8—figure supplement 2*). In these cells, the duration of BF-induced inhibition, as measured as the mean delay between the flash and the first spike after the flash, varied across sweeps, making this estimate imprecise (*Figure 8—figure supplement 2A*).

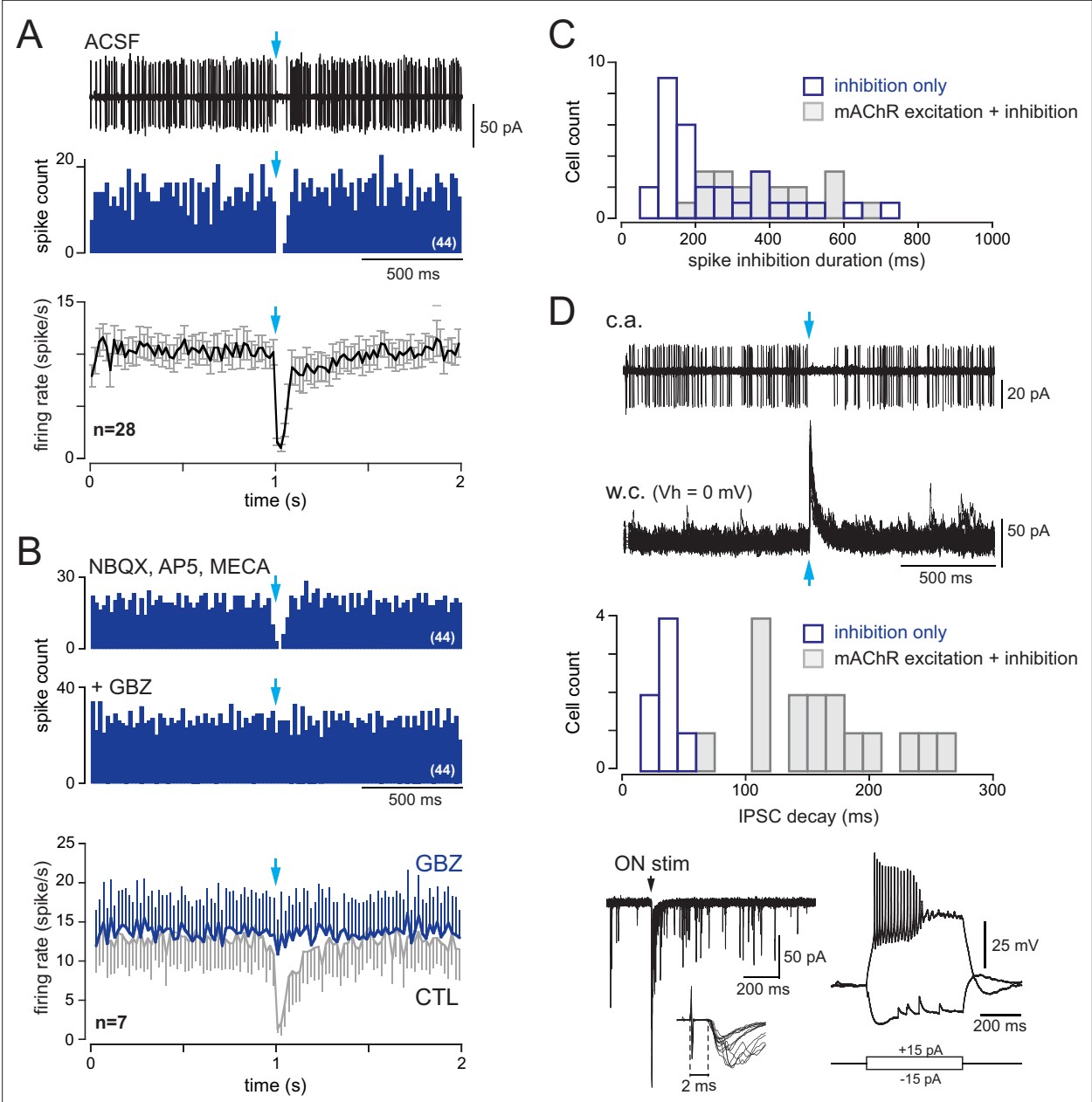

**Figure 8.** Type 2.2 periglomerular (PG) cells are inhibited by the basal forebrain (BF) GABAergic input. (**A**) 10 superimposed loose cell-attached (LCA) traces and the corresponding peri stimulus time histogram (PSTH) for 44 consecutive trials (bin 20 ms) in a cell from a dlx5/6 mouse. Photostimulation of the BF fibers (blue arrow) transiently blocked spiking. Bottom: average firing frequency per bin (20 ms) and per episode for 28 cells that were inhibited by the BF input, without evidence of parallel cholinergic excitation (**Figure 8—figure supplement 1**). (**B**) BF-induced spiking inhibition persisted in the presence of 6-nitro-7-sulfamoylbenzo[f]quinoxaline-2,3-dione (NBQX), D-2-amino-5-phosphonopentanoic acid (D-AP5), and mecamylamine but was blocked by gabazine (GBZ). The two PSTHs are from the same cell as in (**A**) (bin 20 ms). Bottom: average firing rate per bin (20 ms) and per episode for seven cells in control conditions (gray line, six cells in the presence of blockers, one cell in artificial cerebrospinal fluid [ACSF]) and when GBZ (5 μM) was added (blue). (**C**) Duration of the post-stimulus spiking inhibition in cells with a BF-induced inhibitory response (blue) vs. in cells with a biphasic inhibition-muscarinic excitation response (gray). See also **Figure 8—figure supplement 2** for caveats in these measurements. (**D**) Whole-cell characterization of a cell with an inhibitory response. Top: BF impact on firing (cell-attached recording, 38 episodes are superimposed) and BF-evoked IPSCs (whole-cell recording, 15 superimposed episodes). The histogram compares the decay time constants of photo-evoked GABAergic IPSCs in seven PG cells that were only inhibited (blue bars) vs. in cells with a mixed GABA/ACh response (gray bars). Bottom left: olfactory nerve (ON)-evoked EPSCs (left, four superimposed traces). Onset latencies > 2 ms (inset) are consistent with a plurisynaptic response. Bottom right: current-clamp voltage responses to current steps.

The online version of this article includes the following figure supplement(s) for figure 8:

*Figure 8 continued on next page*

**Figure supplement 1.** Absence of cholinergic excitation in periglomerular (PG) cells showing a transient inhibition of spiking.

**Figure supplement 2.** Caveats for the measurements of spiking inhibition duration.

Cells with BF-induced spiking inhibition and no evidence for cholinergic excitation in the cell-attached configuration were subsequently characterized in the WC mode (n = 7, *Figure 8D*). Their electrical membrane resistance was 1119 ± 789 mOhm. BF axons photostimulation evoked large IPSCs (amplitude range 69–464 pA, mean 170 ± 144 pA) with an average decay time constant of 37 ± 9 ms, which was significantly shorter than in type 2.3 PG cells (p<0.0001, Mann–Whitney rank-sum test). Three cells in which ON-evoked responses were recorded responded with a short burst of EPSCs (duration <150 ms). Onset latencies (2.45 ± 0.23 ms) were consistent with plurisynaptic responses. Firing properties were more heterogeneous. Injection of a depolarizing current step evoked a burst of action potentials followed by a plateau (n = 3 cells, as in the example shown in *Figure 8D*) or sustained firing of action potentials (n = 3, not shown). In 5/6 cells, injection of a hyperpolarizing step caused a voltage sag, suggesting the activation of an Ih current. Altogether, these properties match well with those of type 2.2 PG cells (*Najac et al., 2015*). This suggests that type 2.2 PG cells only receive an inhibitory GABAergic input from the BF.

## GABA is excitatory in a minority of CR-expressing type 2.1 PG cells

Finally, photo-evoked BF inputs elicited a single spike (*Figure 9A*) or, more rarely, a doublet (*Figure 9C*) in a small number of cells in dlx5/6 mice (n = 29/383, 7% of the cells tested). These excitatory responses were not seen in ChAT mice, persisted in the presence of NBQX, D-AP5, and mecamylamine (n = 6), and were blocked by gabazine (n = 4, *Figure 9B and D*), suggesting that they were caused by a direct excitatory GABAergic input. It would not be surprising if GABA was depolarizing in CR-expressing PG cells that retain many properties of newborn immature neurons (*Benito et al., 2018*). Consistent with this hypothesis, most of the cells with an excitatory response (26/29) had no or little (<1 Hz) spontaneous firing activity, similar to CR + PG cells (*Benito et al., 2018*; *Fogli Iseppe et al., 2016*). However, photo-evoked action potentials occurred within a short delay after the flash, as expected for synaptically driven spikes, in only 15/29 cells (average spike timing 13.4 ± 9.9 ms, *Figure 9A and E*). In the other cells, the average delay was longer (329 ± 191 ms) and more variable (n = 14, *Figure 9C and E*), suggesting that distinct mechanisms drive the two types of response. Strikingly, the photo-evoked BF synaptic input evoked a small but detectable gabazine-sensitive capacitive current that was inward in 14/15 cells responding with early spikes (*Figure 9A, B, and E*), whereas it was outward in all the cells responding with delayed spikes (*Figure 9C–E*). Thus, a likely explanation of the data is that a depolarizing GABAergic input directly triggered early spikes, whereas delayed spikes could be induced by rebound depolarization following a hyperpolarizing GABAergic IPSP, as often seen in CR + PG cells (*Benito et al., 2018*). Eight cells responding with early action potentials in the cell-attached mode (average timing 20 ± 19 ms) were subsequently characterized in the WC configuration. All of them had properties consistent with those of CR + PG cells, that is, a large input resistance (6 ± 3.1 GΩ), a characteristic voltage response to depolarizing current steps, and a large and fast photo-evoked BF IPSC (amplitude range 133–1300 pA, mean 370 ± 139 pA; weighted decay time constant 7.7 ± 3.6 ms) (*Figure 9G*).

The BF input induced a detectable capacitive current, but no spike, in 70 additional cells, all with no or little spontaneous activity (these cells were classified as nonresponsive in *Figure 3—figure supplement 2*). BF-evoked capacitive current was inward in 54% of these cells (n = 38/70) and outward in the other cells (n = 32, *Figure 9F*). Eight of these cells were characterized with WC recording, and all displayed the typical intrinsic and synaptic properties of CR + PG cells including a high membrane resistance (3 ± 1.2 GΩ) and large and fast BF IPSCs (amplitude: 284 ± 107 pA; decay 12.2 ± 2.9 ms). Altogether, this analysis suggests that the BF GABAergic input is depolarizing in as much as half of the CR-expressing type 2.1 PG cells that may have an elevated chloride reversal potential, more depolarized than the membrane potential. However, this input drives spiking in only a minority of them.

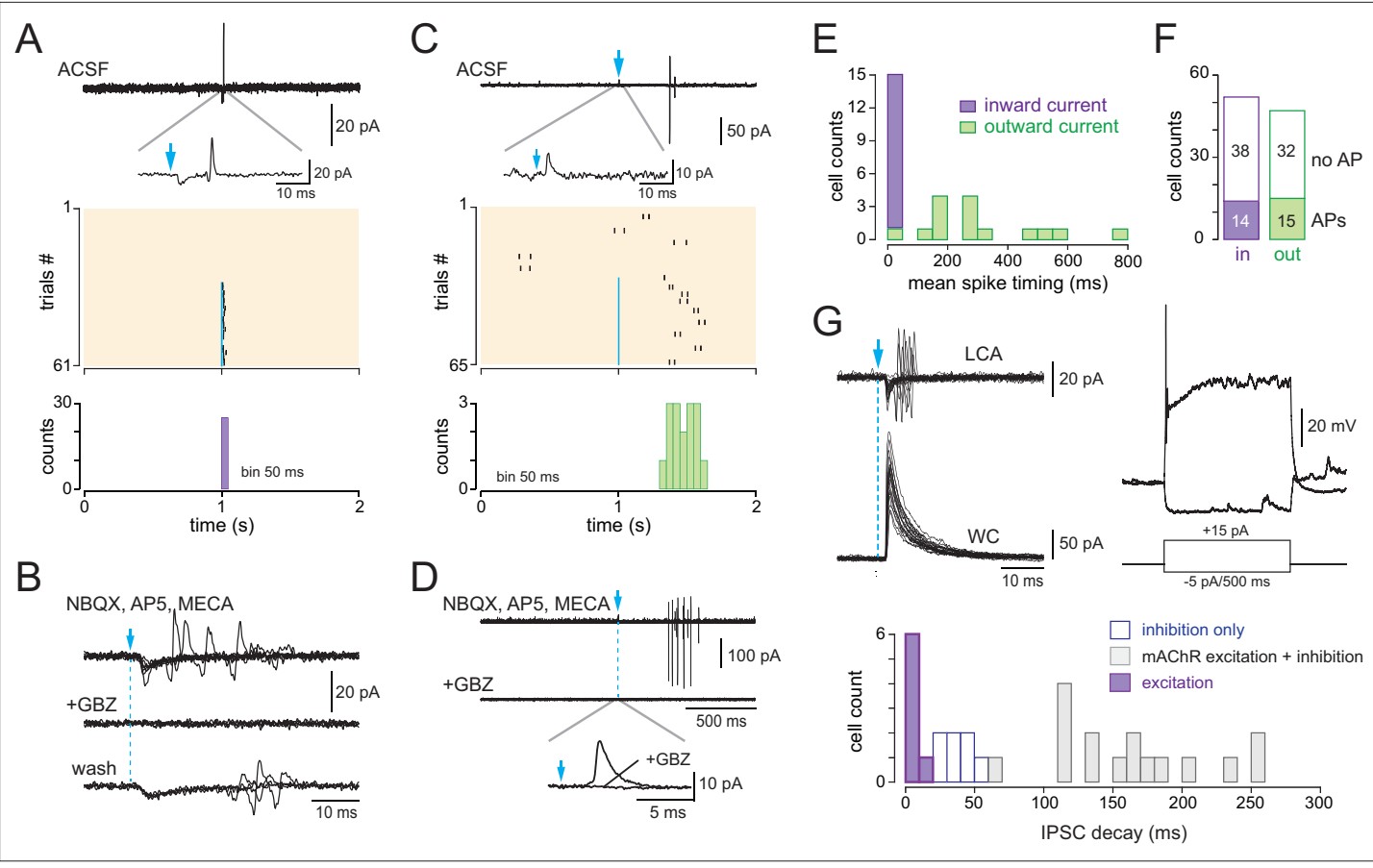

**Figure 9.** Basal forebrain (BF)-evoked GABAergic excitation in a fraction of type 2.1 periglomerular (PG) cells. (**A**) Example of a BF-evoked excitatory response in a PG cell. One representative loose cell-attached (LCA) recording episode (duration 2 s) is shown on top. Photostimulation of the BF fibers (blue arrow) induced an inward current followed by a single spike (inset). Each tick is a spike in the raster plot of these cell responses (stimulation: episodes 31–61). Bottom: corresponding peri stimulus time histogram (PSTH). Note the temporal precision of the evoked spikes. Bin size 50 ms. (**B**) Same cell as in (**A**). The evoked inward current and the evoked spike both persisted in 6-nitro-7-sulfamoylbenzo[f]quinoxaline-2,3-dione (NBQX), D-2-amino-5-phosphonopentanoic acid (D-AP5), and mecamylamine but were blocked by gabazine. At least four traces are superimposed in each condition. (**C**) Another example of a PG cell responding to the photostimulation with action potentials. Photostimulation (blue arrow, episodes 31–65 in the raster plot) induced an outward current (inset) and, in some trials, a delayed doublet of spikes. Bottom: corresponding cumulative PSTH (bin size 50 ms). (**D**) Same cell as in (**C**). Gabazine blocked both the evoked outward current and the evoked spikes. (**E**) Distribution histogram of the average spike timing in each cell responding with spikes. Cells in which the stimulation evoked an inward capacitive current are in violet, and cells in which the stimulation evoked an outward current are in green. (**F**) Total number of cells responding with an inward (violet) or an outward (green) capacitive current followed or not with evoked spikes. (**G**) Whole-cell characterization of a cell excited by the BF input. Left: BF-evoked spike response (LCA recording, top, 20 consecutive episodes are superimposed) and BF-evoked IPSC (whole-cell recording, 20 consecutive responses). Right: current-clamp voltage responses to current steps in the same cell. The distribution histogram compares the decay time constants of photo-evoked GABAergic IPSCs in the seven PG cells that were excited (violet bars) vs. in type 2.2 cells with an inhibitory response (blue) and type 2.3 cells with a mixed GABA/ACh response (gray bars).

## Discussion

This study shows that OB-projecting BF neurons have diverse impacts on PG cells. GABAergic inputs potently block the discharge of type 2.2 and type 2.3 PG cells with a target-specific time course. In contrast, GABA release is excitatory and eventually triggers action potentials in a fraction of type 2.1 PG cells. Data also reveal that BF cholinergic fibers strongly and exclusively excite type 2.3 PG cells. Thus, intraglomerular inhibition of principal neurons mediated by PG cells can be modulated in various ways by multiple BF pathways that potentially regulate olfactory processing in a context and behavior-specific manner.

## A previously ignored cholinergic pathway

The main finding of this study is that endogenous phasic ACh release from BF cholinergic neurons selectively evokes a remarkably strong and reliable muscarinic excitation in a previously overlooked PG cell subtype, which are referred to as type 2.3 PG cells. This novel cholinergic pathway concerns a small population of neurons lacking a molecular marker and evokes small muscarinic EPSP/EPSCs that rapidly runs down in WC recording, explaining why it has been missed until now. Here, I provide pharmacological evidence suggesting that the slow cholinergic response is mediated by M1 mAChRs that suppress an M current. This downstream mechanism classically washes out quickly, and future experiments using perforated patch-clamp experiments will be necessary to confirm this hypothesis.

This pathway adds to other mAChR-dependent mechanisms capable of increasing tonic inhibition in mitral and tufted cells. Activation of M1 receptors increases the excitability of granule cells, the most abundant interneurons in the OB, by potentiating current-evoked afterdepolarization (*Pressler et al., 2007*). mAChR activation also directly enhances transmitter release at reciprocal dendrodendritic synapses between mitral and granule cells (*Castillo et al., 1999*; *Ghatpande and Gelperin, 2009*) or between juxtaglomerular interneurons and mitral/tufted cells (*Liu et al., 2015*). However, there is yet no evidence that endogenous ACh can recruit these previously described pathways. In the previous studies, ACh or cholinergic agonists were exogenously applied on slices. This results in prolonged and uniform activation of synaptic and extrasynaptic ACh receptors and induces multiple concomitant effects that may not necessarily be evoked by physiological release of ACh, even in case of strong afferent activity in cholinergic neurons leading to diffuse volume transmission (*Unal et al., 2015*).

## Physiological implications

Like other sensory systems, olfactory perception is context-dependent. This modulation already takes place in OB circuits, where odor-evoked neural responses depend on reward (*Doucette and Restrepo, 2008*) or on the difficulty of the task (*Koldaeva et al., 2019*) and are shaped by learning and experience (*Martin et al., 2004*; *Chu et al., 2017*; *Ross and Fletcher, 2018*). Whether BF cholinergic innervation of the OB plays a role in context-dependent neuromodulation is suspected, but has never been proven.

In the cortex, transient increased cholinergic signaling signals a transition to a behaviorally important context and adjust neural output to improve task performances (*Gritton et al., 2016*; *Kuchibhotla et al., 2017*; *Pinto et al., 2013*). In Pavlovian learning paradigms, BF cholinergic neurons respond with brief and temporally precise burst of activity to reward or aversive stimuli and to conditioning stimuli, including olfactory cues (*Guo et al., 2019*; *Hangya et al., 2015*; *Hanson et al., 2021*; *Crouse et al., 2020*). This results in fast and precise transients of ACh in target sensory cortical areas (*Guo et al., 2019*) or in the basolateral amygdala (*Crouse et al., 2020*). The physiological dynamics of ACh within the OB are unknown, but it is tempting to speculate that similar ACh transients are evoked in the OB during olfactory-guided aversive or appetitive learning, a behavior that critically depends on M1 mAChRs in the OB (*Ross et al., 2019*).

I showed that a single stimulation of the cholinergic BF axons, which evokes transient, temporally, and spatially precise release of ACh, triggers a long-lasting discharge in type 2.3 PG cells. This target-specific muscarinic response likely involves synaptic or perisynaptic mAChRs and provides support for phasic, spatially restricted cholinergic transmission as opposed to spatially diffuse volume transmission (Sarter and Lustig, 2020). The same stimulus repeated every 2 s, a low-frequency stimulation that is insufficiently strong to induce massive diffusion of ACh in the extracellular space, rapidly transforms type 2.3 PG cells that usually fire at low rate into tonically active neurons that fire at high frequency. This, in turn, leads to a rapid and persistent increase of synaptic inhibitory currents in principal neurons, thus potentially driving OB circuits in a different state of activity. This target-specific muscarinic transmission may have widespread circuit implications because single cholinergic axons frequently ramify and innervate multiple glomeruli in different OB areas (*Hamamoto et al., 2017*). This could ultimately reduce the firing rate in output neurons but could also shape the temporal structure of mitral and tufted cells output at diverse time scales. Inhibitory inputs regulate spike timing and synchrony in mitral and tufted cells (*Najac et al., 2015*; *Schoppa, 2006*; *Shao et al., 2012*), and inhibition of eTCs might modulate slow glomerulus-specific coordinated activity (*De Saint Jan et al., 2009*; *Hayar et al., 2004*; *Najac et al., 2011*). Based on previous studies on ACh functions, increased

inhibition driven by the muscarinic excitation of type 2.3 PG cells may improve olfactory perception of behaviorally important odorants. Interestingly, BF GABAergic fibers innervating type 2.3 PG cells potently block their activity and could act as a powerful brake to reverse the cholinergic effects. Future in vivo experiments will be necessary to explore these possibilities.

In vivo, optogenetic stimulation of the cholinergic axons in the OB of *Chat*^Cre mice increases mitral and tufted cells' spontaneous and odor-evoked firing (*Böhm et al., 2020*; *Rothermel et al., 2014*). This result seems at odds with the expected implications of the new muscarinic pathway described in this study. However, photostimulation was strong and sustained in these in vivo studies (light continuously on for 10 s). Although it is difficult to compare stimulations in vivo and in slices, a prolonged photostimulus could recruit additional cholinergic pathways that need volume transmission to be activated and that have opposite impacts compared with those of type 2.3 PG cells. For instance, TH-expressing dopaminergic/GABAergic juxtaglomerular neurons express the M2 mAChR (*Crespo et al., 2000*; *Hamamoto et al., 2017*). Activation of these receptors inhibits tonically active TH-expressing cells (*Pignatelli and Belluzzi, 2008*) that provide an inhibitory drive to mitral and tufted cells (*Liu et al., 2016*; *Whitesell et al., 2013*; *Zhou et al., 2020*). Thus, understanding the physiological impact of the muscarinic excitation of type 2.3 PG cells on the OB output and network activity will require targeted stimulations that selectively engage this pathway in vivo as well as specific approaches to determine the in vivo activity of the cholinergic afferents.

## Multiple cell-type-specific pathways for BF control of glomerular inhibition

Results of this study also provide new insights into PG cell diversity. Immunohistochemical studies have already demonstrated that the few classical markers commonly used to label PG cells do not label all of them (*Panzanelli et al., 2007*; *Parrish-Aungst et al., 2007*; *Whitman and Greer, 2007*). Yet, most functional studies only distinguish type 1 and type 2 PG cells and ignore type 2's diversity. CR-expressing type 2.1 PG cells are by far the most abundant, representing 40–50% of the entire PG cell population. They are predominately generated postnatally and persist in an immature stage in terms of connectivity and membrane properties (*Benito et al., 2018*). The present data suggest that BF GABAergic inputs are excitatory in a fraction of them, likely the most immature, whereas GABA is inhibitory in the other PG cells. The functional impact of this excitation is unclear because it concerns a minority of PG cells and there is no evidence that immature CR + PG cells form output synapses. However, this result may explain why GABA appears predominantly excitatory in calcium imaging of unidentified PG cells (*Parsa et al., 2015*). Type 2.3 PG cells constitute about 20% of the whole PG cell population and are approximately as numerous as type 2.2 PG cells (*Sanz Diez et al., 2019*). These regular spiking interneurons have pluri-synaptic long-lasting ON-evoked responses and receive remarkably slow BF IPSCs that readily distinguish them from type 2.1 and type 2.2 PG cells. As shown here, their muscarinic input is another selective feature. Although their input and output connections are not firmly established, type 2.3 PG cells are presumably activated by the glutamate released from mitral and tufted cell dendrites and most likely release GABA unselectively onto mitral and tufted cells. Consistent with this idea, a previous study showed that mAChR activation within glomeruli increases IPSCs equally well in mitral and tufted cells (*Liu et al., 2015*). The present data confirm that type 2.3 PG cells inhibit various classes of tufted cells, but I found no evidence that they also inhibit mitral cells. However, this negative result has to be interpreted with caution as it is challenging in slices from adult mice to find mitral cells projecting in surface glomeruli, a technical requirement for optimal LED stimulation of the cholinergic afferents.

The functional implications of PG cells' diversity are not known. Cholinergic and GABAergic inputs from the BF may provide physiological tools to manipulate each PG cell subtype selectively in future studies exploring this question. Like elsewhere in the brain, BF GABAergic neurons are highly diverse and each cell population makes cell-type-specific long-range connections (*Do et al., 2016*) and plays specific functions. For instance, somatostatin (SOM)- and parvalbumin (PV)-expressing subpopulations have distinct impacts on arousal control (*Anaclet et al., 2018*; *Xu et al., 2015*) or food intake (*Zhu et al., 2017*). Distinct classes of BF GABAergic neurons may thus modulate distinct classes of PG cells. Similarly, muscarinic excitation of type 2.3 PG cells may involve a specific population of BF cholinergic neurons. There are at least two distinct types of BF cholinergic neurons that differ in their firing modes and synchronization properties and that are differently engaged during behaviors (*Laszlovszky et al.,*

*2020*). This specificity could also rely on connectivity. For instance, BF cholinergic neurons modulating distinct areas are driven by specific combinations of synaptic inputs (*Do et al., 2016*; *Gielow and Zaborszky, 2017*; *Zaborszky et al., 2015*; *Zheng et al., 2018*). The recent discovery of a genetically defined subpopulation of cholinergic neurons that selectively innervates a specific subgroup of deep short-axon cells in the OB (*Case et al., 2017*) also supports the hypothesis of cell-specific innervation of OB interneurons by specific subsets of cholinergic neurons. Hence, each of the BF neuromodulatory pathways innervating PG cells might be independently recruited during specific tasks or internal states.

## Materials and methods

### Animals and ethical approval

All experimental procedures were approved by the French Ministry and the local ethic committee for animal experimentation (CREMEAS; agreement number/reference protocol: APAF-IS#5250–2016042115058488v3 and v7). Mice were housed in the animal facility (Chronobiotron, UMS3415, CNRS, University of Strasbourg) with ad libitum access to food and water in accordance with the European Convention 2010/63/EU on the protection of animals used for scientific purposes. Adult heterozygous *dlx5/6*[Cre] mice (n = 36, 34 females and 2 males, C57BL6/J background; Jackson Laboratory stock no: 008199; *Monory et al., 2006*) and *Chat*[Cre] mice (n = 44 of either sex, CD1 background; Jackson Laboratory stock no: 006410; *Rossi et al., 2011*) were used in this study.

### Stereotaxic viral injection

3–8-week-old mice were anesthetized with intraperitoneal injection of Zoletil 50 (tiletamine/zolazepam, 60–70 mg/kg) and Rompun 2% (xylasine, 18–20 mg/kg) and placed in a stereotaxic apparatus. Metacam (meloxicam, 2 mg/kg, SC injection) and lurocaine + bupivacaine (2 mg/kg both, SC, local) were administered prior incision. Mice were craneotomized, and a volume of 300–400 nl of AAV9.EF1a.DIO.hChR2(H134R).eYFP.WPRE.hGH was stereotaxically injected in the left hemisphere at 0/+0.2 mm AP, 1.4/1.6 mm ML, and 5.4/5.6 mm DV from bregma. Viruses were purchased from the University of Pennsylvania Viral Vector Core (RRID:Addgene#20298; virus titer $1.8 \times 10^{13}$ vg/ml) and the Canadian Neurophotonics Platform Viral Vector Core Facility (RRID:SCR_016477, virus titer $9 \times 10^{12}$ GC/ml). After surgery, antisedan (atipamezol, 2.5%) was injected IP and mice were rehydrated with 0.5 ml of NaCl 0.9% and placed under a heating lamp. Mice recovered during 2–4 weeks after injection before anatomical or physiological experiments.

### Slice preparation

Mice were killed by cervical dislocation and the OB rapidly removed in ice-cold oxygenated (95% $O_2$–5% $CO_2$) cutting solution containing (in mM) 83 NaCl, 26.2 $NaHCO_3$, 1 $NaH_2PO_4$, 2.5 KCl, 3.3 $MgSO_4$, 0.5 $CaCl_2$, 70 sucrose, and 22 D-glucose (pH 7.3, osmolarity 300 mOsm/l). Horizontal OB slices (300-µm-thick) were cut using a Microm HM 650V vibratome (Microm, Germany) in the same solution, incubated for 30–40 min at 34°C, stored at room temperature (RT) in a regular ACSF until use. ACSF contained (in mM) 125 NaCl, 25 $NaHCO_3$, 2.5 KCl, 1.25 $NaH_2PO_4$, 1 $MgCl_2$, 2 $CaCl_2$, and 25 D-glucose and was continuously bubbled with 95% $O_2$–5% $CO_2$.

### Electrophysiological recordings

Slices were transferred to a recording chamber and perfused with ACSF at 32–34°C under an upright microscope (SliceScope, Scientifica, Uckfield, UK) with differential interference contrast (DIC) and fluorescence optics. Spontaneous action potential firing activity was monitored using LCA recording (15–100 MΩ seal resistance). LCA recordings were made in the voltage-clamp mode of the amplifier (multiclamp 700B, Molecular Devices, Sunnyvale, CA) with no current injected through the pipette. In these conditions, large and fast membrane potential changes such as action potentials are detected as capacitive currents flowing across the patch capacitance (*Barbour and Isope, 2000*). A regular patch pipette filled with ACSF was used on several successively recorded cells. WC PG cell recordings were made with glass pipettes (4–7 MΩ) filled with a regular K-gluconate-based internal solution containing (in mM) 135 K-gluconate, 2 $MgCl_2$, 0.025 $CaCl_2$, 1 EGTA, 4 Na-ATP, 0.5 Na-GTP, and 10 HEPES (pH 7.3, 280 mOsm, 15 mV junction potential). The intracellular solution used to record the muscarinic EPSC

was adapted from *Lawrence et al., 2006* and contained (in mM) 110 K-gluconate, 4 MgCl$_2$, 0.1 EGTA, 4 Na$_2$-ATP, 0.5 Na$_2$-GTP, 10 HEPES, and 10 phosphocreatine (pH 7.3, 250 mOsm/l, 15 mV junction potential). WC voltage-clamp recording from mitral and tufted cells was made with an internal solution containing (in mM) 120 Cs-MeSO$_3$, 20 tetraethylammonium-Cl, 5 4-aminopyridine, 2 MgCl$_2$, 0.025 CaCl$_2$, 1 EGTA, 4 Na-ATP, 0.5 Na-GTP, and 10 HEPES (pH 7.3, 280 Osm/l, 10 mV junction potential). Atto 594 (10 µM, Sigma) was systematically added to the internal solution in order to visualize the cell morphology during the recording. Optical stimulation of the BF axons was done using a blue LED (490 nm, pE 100, CoolLED Ltd., Andover, UK) directed through the ×40 objective of the microscope at 50–100% of its maximum power (5 mW at the objective output) and driven by the AxoGraph X acquisition software (AxoGraph Scientific). ONs projecting inside a given glomerulus were electrically stimulated using a theta pipette filled with ACSF. The electrical stimulus (100 µs) was delivered using a Digitimer DS3 (Digitimer, Welwyn Garden City, UK). Recordings were low-pass-filtered at 2–4 kHz and digitized at 20 kHz using the AxoGraph X software. In WC voltage-clamp recordings, access resistance was not compensated. Voltages indicated in the article were corrected for the junction potential.

## Cell selection

PG cells were selected based on the small size of their cell body and their position within the first rings of cells surrounding the glomerulus. Although they have a larger cell body, it cannot be excluded that TH-expressing cells or eTCs have been erroneously included in the LCA recording dataset. However, these cells usually have remarkable spontaneous activity patterns (eTCs are rhythmically bursting, TH+ cells have a highly regular rhythmic discharge) and cells with this kind of activity constituted a minority of the dataset. eTC, superficial or middle tufted cells (collectively called s/mTC), and mitral cells were identified based on the localization of their soma and the presence or not of lateral dendrites in the external plexiform layer, as seen during WC recording by visual inspection of the dye-filled cell morphology. In addition, some cells were filled with biocytin for post-hoc anatomical reconstruction. Thus, eTCs were selected based on their large pear-shaped soma within the glomerular layer, a short and thick apical dendrite extensively ramifying into a single glomerulus and the lack of lateral dendrites. In addition, they often, but not always, spontaneously fired short bursts of action potentials in the cell-attached configuration even in the presence of NBQX, D-AP5, and mecamylamine. Superficial tufted cells were found at the border between the glomerular layer and the external plexiform layer. Compared to eTC, their soma was located further from the glomerulus into which they projected, and they had long lateral dendrites extending into the external plexiform layer. Middle tufted cells and mitral cells had large cell bodies located in the external plexiform layer and the mitral cell layer, respectively, a thick apical dendrite projecting into a single glomerulus and long lateral dendrites in the external plexiform layer.

## Morphological reconstruction

Neurobiotin or biocytin (Vector Laboratories Inc, Burlingame, CA) was added to the intracellular solution (1 mg/ml). The patch pipette was slowly retracted after the recording to avoid damaging the cell body. The slice was then fixed in 4% paraformaldehyde (PFA) overnight, washed three times in PBS, and incubated in a permeabilizing solution containing Alexa Fluor 555-conjugated streptavidin (1 µg/ml; Thermo Fisher Scientific, Waltham, MA) overnight. After three rinses with PBS, sections were mounted in Vectashield Hardset with DAPI (Vector Laboratories, Inc). Labeled cells were imaged with a confocal microscope (Leica TCS SP5 II).

## Immunohistochemistry

ChAT mice expressing ChR2-eYFP in BF neurons were deeply anesthetized with zolazepam tiletamine/xylasine and transcardially perfused with PBS at RT followed by 4% PFA (4°C). Brains were removed, postfixed 3–6 hr in 4% PFA at 4°C, rinsed in PBS, and incubated in PBS until cut on a vibratome (VT 1000S, Leica). 50-µm-thick coronal sections were collected and stored in PBS. For ChAT staining, sections were incubated overnight at 4°C with a goat anti-ChAT (1:1000; Millipore, Cat# AB144P, RRID:AB_2079751) in Tris-Triton buffer containing 2% donkey serum and 0.2% Triton X100. After three washes in Tris-Triton, sections were incubated for 1 hr at RT with Alexa Fluor 647-conjugated donkey anti-goat (1:500; Thermo Fisher Scientific, Cat# A-21447, RRID:AB_2535864). After three washes, sections were mounted in Prolong Diamond Antifade Mountant (Thermo Fisher) or Vectashield

Hardset with DAPI. Images were taken using a Leica TCS SP5 II confocal microscope or a Zeiss Axio Imager M2 for mosaic images. Immunostained and EYFP-expressing cells were manually counted using the cell counter plug-in on Fiji software.

## Fluorescence in situ hybridization

FISH, combined with immunohistochemistry, was done using the RNAscope Multiplex Fluorescent Reagent Kit v2 (Advanced Cell Diagnostics Cat# 323100) according to the manufacturer's protocols. Briefly, whole brains from 6-week-old transcardially perfused mice (n = 3, male and female wild-type littermates from the *dlx5/6*<sup>Cre</sup> colony) were extracted and immediately placed in 4% PFA, post-fixed 6 hr in 4% PFA at 4°C, and cryoprotected with successive incubations in 10, 20, and 30% sucrose solution. OBs were then embedded in Tissue-Tek OCT, frozen on dry ice, and stored at –80°C until sliced with a cryostat (Leica CM3050 S) into 10–15 µm coronal sections, adhered to SuperFrost Plus slides (VWR), and immediately refrozen at –80°C.

Unless otherwise stated, the probe and all reagents were provided in the RNAscope Multiplex Fluorescent Reagent Kit v2. On day 1, samples were washed in PBS to remove OCT, incubated 30 min at 60°C, post-fixed in 4% PFA for 15 min at 4°C, and dehydrated with 50 (x1), 70 (x1), and 100% (x2) ethanol washes for 5 min. Slides were air-dried and a barrier drawn around the tissue section with an Immedge hydrophobic barrier pen (Vector Laboratories Inc). Endogenous peroxidase activity was blocked using hydrogen peroxide for 10 min at RT. Sections then underwent antigen retrieval by submersion into boiling (~98–102°C) co-detection target retrieval solution for 5 min and were rinsed in distilled water (five times) and then in PBS-tween (PBS-T, 1 time). Sections were incubated overnight at 4°C with primary antibodies (rabbit anti-CR, 1:1000; Swant Cat# 7697, RRID:AB_2721226, or rabbit anti-TH, 1:500, Millipore, Cat# AB152, RRID:AB_390204) diluted in the co-detection antibody diluent provided by the manufacturer.

On day 2, slices were washed in PBS-T three times for 2 min, post-fixed in PFA 4% for 30 min at RT, washed four times in PBS-T, and treated with protease reagent for 30 min at 40°C. After rinsing twice in distilled water, sections were incubated with the RNAscope probe for Chrm1 *(RNAscope Probe- Mm-Chrm1, target region 851-1994; Cat# 495291-C1)* for 2 hr at 40°C for the hybridization step. Sections were then washed twice in wash buffer at RT, then incubated in Multiplex v2 AMP 1 (40°C for 30 min), AMP 2 (40°C for 30 min), and AMP3 (40 °C for 15 min) for the amplification steps. After rinsing twice in wash buffer, sections were incubated in HRP-C1 for 15 min at 40°C, washed twice in wash buffer, and incubated 30 min at 40°C with the Opal 570 fluorophore (Akoya Biosciences, Cat# FP1488001KT) to mark Chrm1. Sections were washed and subsequently incubated in HRP blocker for 15 min at 40°C. The sections were then further incubated in a secondary antibody (Alexa Fluor 488 goat anti-rabbit; 1:500; Thermo Fisher Scientific, Cat# A-11008, RRID:AB_143165) diluted in co-detection antibody diluent for 45 min at RT to visualize CR or TH. The slides were then washed in PBS-T, incubated with DAPI for 30 s at RT, and covered with ProLong gold antifade mounting medium (Thermo Fisher Scientific, RRID:SCR_015961).

Confocal images were acquired using a Leica TCS SP5 II confocal microscope through a ×20 air objective and a ×63 oil-immersion objective. Images were analyzed using the Fiji software. Hybridization signal was converted into a binary mask by setting a single threshold. Hybridization dots were counted using the analyze particle function on Fiji software, and DAPI-stained cells were manually counted using the cell counter plug-in. Normalized fluorescence intensity profiles were estimated within a 650 × 50 µm area spanning from the nerve layer to the granule cell layer and centered on the mitral cell layer.

## Drugs

Chemicals used to prepare cutting, recording, and internal solutions were acquired from MilliporeSigma, Carl Roth, and Fisher Scientific. NBQX, D-AP5, 2-(3-carboxypropyl)-3-amino-6-(4 methoxyphenyl)pyridazinium bromide (gabazine), mecamylamine hydrochloride, and atropine were purchased from Abcam Biochemicals. Scopolamine hydrobromide, pirenzepine, and XE-991 were purchased from Tocris Bioscience.

## Electrophysiological analysis

Action potential capacitive currents were automatically detected by the AxoGraph X software using an amplitude threshold. The timing of each spike was used to construct peri stimulus time histograms

(PSTHs) representing the total number of action potentials per time period (bin of 20 ms for 2-s-long episodes, of 200 ms for 15 s or longer episodes) across several consecutive sweeps (>30 for 2-s-long episodes, >10 for 15-s-long episodes). For cell classification, a post-stimulus spiking inhibition was a statistically significant (paired $t$-test or paired Wilcoxon signed-rank-sum test) decrease in spike rate within a 100-ms or 200-ms-long time period immediately after the flash compared to the same period preceding the flash across at least 30 consecutive trials. The duration of spiking inhibition was calculated as the average duration between the flash and the first spike fired after the flash. A muscarinic excitation was a statistically significant increase in spike probability within a 1 s period starting 1 s after the flash compared to the 1 s period immediately preceding the flash across at least 10 consecutive trials.

Photo-evoked IPSC amplitudes were measured as the peak of an average response computed from multiple sweeps. The decay of photo-evoked IPSCs was most often best fitted with a double exponential with time t = 0 at the peak of the current. Time constant values indicated in the text are weighted decay time constants calculated using the following equation: $\tau_w = (\tau_1 A_1 + \tau_2 A_2)/(A_1 + A_2)$, where $\tau_1$ and $\tau_2$ are the fast and slow decay time constants, and $A_1$ and $A_2$ are the equivalent amplitude weighting factors.

EPSCs and IPSCs were automatically detected by the AxoGraph X software using a sliding template function. The onset of ON-evoked EPSCs was measured at 5% of the first peak of the response. The latency of an ON-evoked EPSC was defined as the time interval between the beginning of the stimulation artifact and the onset of the first EPSC. To estimate the duration of an ON-evoked plurisynaptic excitatory response, PSTHs representing the cumulative number of EPSCs per 20 ms bin across several consecutive sweeps were constructed. The time needed after stimulation for the EPSC frequency to come back to baseline frequency +2 SD during at least five consecutive bins was then determined. Baseline frequency was calculated over the 25 bins (i.e., 500 ms) preceding the stimulation.

Data are presented as mean ± SD in the text and as mean ± SEM in the graphs for display purpose. Data points from experiments were tested for normality using a Shapiro–Wilk test. Experiments with a normal distribution were tested for statistical significance with a paired Student's $t$-test. Experiments with skewed distributions were tested for statistical significance using a paired Wilcoxon signed-rank-sum test. For experiments comparing data points from different cells, statistical significance was determined using an unpaired $t$-test (normal distribution) or a Mann–Whitney test (non-normal distribution).

## Acknowledgements

This work was supported by the Centre National de la Recherche Scientifique (UPR3212) and the Université de Strasbourg (UPR3212). I thank Alvaro Sanz Diez, Julie Perraud, and Andréa Grinner, who contributed preliminary data. Ipek Yalcin and Matilde Cordero-Erausquin (Institut des Neurosciences Cellulaires et Intégratives, Strasbourg) for the kind gift of the *dlx5/6*Cre mice and *Chat*Cre mice. Pierre Hener and Jennifer Kaufling (Institut des Neurosciences Cellulaires et Intégratives, Strasbourg) for their technical support with the RNAscope assay. Marion Najac, Philippe Isope, Antoine Valera, Karin Aubrey, and Yo Otsu for their helpful comments on the manuscript. I also thank Dr. Sophie Reibel-Foisset and the staff of the animal facility (Chronobiotron, UMS 3415 CNRS and Strasbourg University) for technical assistance and members of the team Physiology of Neural Networks for their support throughout the project.

## Additional information

### Funding

| Funder | Grant reference number | Author |
| --- | --- | --- |
| Université de Strasbourg | UPR3212 | Didier De Saint Jan |
| Centre National de la Recherche Scientifique | UPR3212 | Didier De Saint Jan |

| Funder | Grant reference number | Author |
|---|---|---|

The funders had no role in study design, data collection and interpretation, or the decision to submit the work for publication.

## Author contributions
Didier De Saint Jan, Conceptualization, Data curation, Investigation, Methodology, Writing - original draft

## Author ORCIDs
Didier De Saint Jan (ID) http://orcid.org/0000-0002-2459-9703

## Ethics
All experiments procedures were approved by the French Ministry and by the local ethic committee for animal experimentation (CREMEAS) (authorization number APAFIS#5250-2016042115058488v3 and v7) . Mice were housed in the animal facility with ad libitum access to food and water. Animals were sacrificed by cervical dislocation following the methods approved by the directive 2010/63/EU of the European Parliament and Council. Surgeries were performed under anesthesia and every effort was made to minimize suffering.

## Decision letter and Author response
Decision letter https://doi.org/10.7554/eLife.71965.sa1
Author response https://doi.org/10.7554/eLife.71965.sa2

# Additional files

## Supplementary files
• Transparent reporting form

## Data availability
All numerical data used to construct graphs in each figure are available on ZENODO, https://doi.org/10.5281/zenodo.6259698https://doi.org/10.5281/zenodo.6259698.

The following dataset was generated:

| Author(s) | Year | Dataset title | Dataset URL | Database and Identifier |
|---|---|---|---|---|
| De Saint Jan D | 2022 | Target-specific control of olfactory bulb periglomerular cells by GABAergic and cholinergic basal forebrain inputs | https://doi.org/10.5281/zenodo.6259698 | Zenodo, 10.5281/zenodo.6259698 |

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
