## [Editor Report]

This study reports on the synaptic impact of basal forebrain stimulation on a population of olfactory bulb interneurons in acute mouse brain slices. The author reveals that optogenetic stimulation of GABAergic basal forebrain afferents by and large inhibits the discharge of periglomerular cells, whereas cholinergic afferents evoke a prolonged, M1 receptor-mediated depolarization and increase in firing in a subpopulation of periglomerular cells. The current study would further our understanding of the olfactory neural circuit and how different neurotransmitters shape postsynaptic neuronal responses.

---

## [Decision Letter]

**Decision letter after peer review:**

Thank you for submitting your article "Target-specific control of olfactory bulb periglomerular cells by GABAergic and cholinergic basal forebrain inputs" for consideration by *eLife*. Your article has been reviewed by 3 peer reviewers, one of whom is a member of our Board of Reviewing Editors, and the evaluation has been overseen by John Huguenard as the Senior Editor. The reviewers have opted to remain anonymous.

All three reviewers agree that the manuscript presented interesting findings on how GABAergic and cholinergic signals impact the input layer of the primary olfactory area of the mouse brain and reports that the effect depends on the cell type. The reviewers appreciate the precise analysis using brain slices and do not have major concerns regarding technical issues. However, the reviewers feel strongly that essential revisions are needed to strengthen the conclusions and to make the paper more accessible to a wide audience.

Essential revisions:

1. The author suggests a classification of type 2 PGs into 3 further subtypes (2.x) based on their expression of CR (2.1), Kv3.1 (2.2) and cholinergic response (2.3). However, it is unclear whether these really reflect a separate cell class. The reviewers suggest that adding further evidence substantiating this claim would be important to avoid creating a potentially unnecessary and confusing nomenclature.

2. A better characterization of the BF neurons stimulated in the experiments with Dlx5/6-cre and Chat-cre mice. Reviewer #2 noted that some technical details regarding mice are missing, and that higher resolution images would, at minimum, be needed. All reviewers suggested a better characterization of transduced neurons: What fraction of Chat neurons are captured in dlx5/6 mice (e.g. assessed by IHC)? Do the Chat-negative neurons labeled in Chat-Cre mice project to OB (perhaps assessed with retrobeads from OB)? Also Dlx5/6-cre mice likely label many GABAergic neurons in the forebrain: Do the author manage to label only cells within the HDB? If not, where else may the GABAergic inputs come from and is it fair to unambiguously refer to them as BF inputs?

3. Reviewer#2 suggested that the "inhibition" of sTC and eTC by cholinergic stimulation be measured in a decrease in firing rate, as opposed to an increase in sIPCS frequency. It is presently unclear that the observed sIPSC effect comes from these cells. While a demonstration of this is likely beyond the scope of this study , showing that M1 receptors are selectively expressed in PG 2,3 cells (see point 1 above, see comments by reviewer #1) or that this effect is also dependent on M-channels would help bridge that gap.

4. Additional clarity on impact and a larger picture. Multiple things can/should be done in this regard:

a. This paper follows another from the same group (Sanz Diez et al. 2019) that already looked at the postsynaptic effects of BF afferents onto OB cells using optogenetics in dlx5/6-cre mice. That paper focused mainly on GABAergic inputs, and reported dense innervation of all layers of the OB and potent GABAergic responses in granule cells, short axon cells and subpopulations of type 2 PG cells (but not type 1s). Reviewers feel that an analysis performed by the same group that ought to be repeated here to compare their new classification scheme with their previous one.

b. This paper builds onto this 2019 paper, but focuses exclusively onto PG cells for both GABA and ACh responses. What motivates this focus is not presently clear, as several classes of OB neurons express nicotinic and muscarinic receptors (Castillo et al. 1999, Liu et al. 2015, Smith et al. 2015, Brunert and Rothermel 2021). A better justification for their exclusive focus on PG cells is warranted, together with some illustration that helps understand how cholinergic excitation and GABAergic inhibition of PG cells might affect OB output.

c. Related to this, the focus of the paper remains vague: is it to define multiple subclasses of type 2 PG cells, in which case the functional GABAergic and cholinergic responses help substantiate the claim that these cells ought to be broken down into 3 subtypes? Or is the focus on the impact of BF inputs to OB function, in which case the focus on type 2 PG cells should be clarified (as well as the identity of the BF cells transduced with ChR2; see point 2)? Can we assume that ACh does not directly influence other cells? And what about the relative contribution of GABAergic signaling onto PG cells vs other interneurons and principal cells?

d. The utility of figure 1 is still unclear. All it does is to confuse the reader as to the potential existence of complex, co-transmission mechanisms, when in fact it derives from describing dlx5/6-cre mice as a specific driver of GABAergic neurons, which it isn't. Dlx5/6 labels neurons that originate in the embryonic ganglionic eminences during development, which includes GABAergic interneurons as well as cholinergic neurons in the forebrain. If one were interested in understanding how ACh modulates OB circuits, as the introduction suggests, one would not use dlx5/6-cre mice to start. Because the subsequent figures use more specific Cre lines, therefore, Figure 1 here may confuse readers who are not aware of the caveats associated with the Dlx5/6cre mice.

*Reviewer #1 (Recommendations for the authors):*

– The paper would be strengthened by converging approaches that add to the data presented. Specifically, M1 AChRs appear to be a molecular identifier of type 2.3 PG neurons. It would be interesting to perform in situ hybridization for M1 AChR to examine the distribution in the olfactory bulb to determine what extent it serves as a marker for this functionally identified class of neurons.

– A novel finding of this paper is that a subset of BF neurons that release only ACh make direct connections with PG 2.3 neurons in the olfactory bulb. However, there is still a substantial population of neurons that co-release both ACh and GABA. In order to more fully appreciate the relative size of the population that releases ACh only, it would be helpful if the author could perform quantification of ChAT+ neurons in the BF that are not labeled by Dlx5/6. This could be achieved by performing ChAT immunohistochemistry in a Dlx5/6 cre mouse that was injected with a cre-dependent virus to estimate the relative size of the co-releasing and non-co-releasing populations.

*Reviewer #2 (Recommendations for the authors):*

Has the author tested the specificity of conditional expression using the AAV? The cre-dependence of conditional expression depends on the production method and the virus titre (Fisher et al. 2019, DOI: 10.1073/pnas.1915974116). If this is available, please describe this in a supplementary figure. If not, at the least, please include the titre information.

Please include more information in the methods about the Cre lines used (stock number, original reference etc).

It would be beneficial if the author could provide high resolution versions of the images used in Figure 1B and Figure 4C in the supplement, to be able to see the labelled structures better.

Figure 1C – if I understand correctly, the "excitation" in the pie chart is sometimes very brief type (as in Figure 1E), and sometimes the longer-lasting type, of the type analysed in Figure 3. It would be better if these could be labelled more precisely.

In general, the definition of cell types is ambiguous and not consistent with the definitions described elsewhere. For example, in Sanz Diez et al., the author used the response latency from the time of ON stimulation as a way to distinguish type 1 vs. type 2 PG cells. Please include analyses that allow comparisons with previous work(s). Similarly, a summary (e.g., a table) that describes the distinguishing properties of the PG cell types (types 2.1, 2.2 and 2.3) would help.

Figure 7 – The effect on sTC and eTCs is an increase in iPSCs in voltage clamp, which is clearly shown. The term "inhibition", however, suggests a cessation of the action potential generation, which is not shown. As the net effect on the overall firing rates would likely depend on many factors, here, it is better to simply call it an increase in inhibitory synaptic inputs or similar.

*Reviewer #3 (Recommendations for the authors):*

– The presentation of the findings could be made stronger by providing a diagram of the OB and its component cells types and microcircuits. This would help non-experts understand which cell types are recorded from and how they fit into the shaping of odor responses in the OB. While the many subtypes of neurons populating the OB, their anatomical connections and functional roles remain incompletely understood, one issue with the current presentation of the findings is that we lose track of the forest for the trees. It is difficult to shake the feeling that we keep slicing smaller and smaller parts of a circuit that is not familiar to many, obscuring the significance of the findings. A better description of the roles that PG cells play in olfaction, especially vis-à-vis other interneurons and principal neurons, supplemented with a clear diagram, is necessary to help emphasize the importance of this work.

– The paper opens with a striking observation: stimulation of BF afferents evokes a range of excitatory and inhibitory responses in PG cells. However, as the paper unfolds and more specific methods targeting BF GABAergic and cholinergic neurons are leveraged, it becomes clear that the inhibitory and excitatory components originate in separate populations of afferents. The striking observation of Figure 1 is therefore nothing more than the result of non-selective axonal stimulation. Is starting with a confusing picture necessary, especially when more specific methods yielding more classical findings are utilized in later figures? Are there reasons to believe that BF GABAergic and cholinergic neurons are co-active in vivo?

– The molecular and synaptic basis for the segregation of GABAergic and cholinergic responses onto different subtypes of PG cells is not investigated. The author interprets this finding as resulting from differences in presynaptic innervation/specialization (e.g. lines 460, or 586-588). While probable, what evidence supports this possibility over others? Can we exclude postsynaptic components, such as differential postsynaptic clustering of GABA receptors, or expression of M1 receptors and/or M-channels? Can volume transmission can be excluded as a mode of release based on the data at hand?

– Figure 9 provides evidence that GABA exerts a direct excitatory influence on some PG cells, possibly because their chloride reversal potential is elevated. The author should provide more evidence to substantiate this claim. For instance, does puffing GABA evoke a train of action potentials?

– Line 149: the author describes that AMPA, NMDA and nicotinic ACh receptor blockers prevent excitation of any interneurons. This is not true if one consideres some muscarinic receptors, and indeed this study goes on to show a striking example of this.

– Line 169: remove 'always', since the rest of the sentence describes exceptions.

– Line 208-210: 89% of ChR2+ neurons in the BF of ChAT-Cre mice are cholinergic, leaving 11% of non-specifically labeled cells. Can the author provide any indication as to which other cells are labeled and whether they also project to OB, potentially contributing to postsynaptic responses?

---

## [Author Response]

Essential revisions:1. The author suggests a classification of type 2 PGs into 3 further subtypes (2.x) based on their expression of CR (2.1), Kv3.1 (2.2) and cholinergic response (2.3). However, it is unclear whether these really reflect a separate cell class. The reviewers suggest that adding further evidence substantiating this claim would be important to avoid creating a potentially unnecessary and confusing nomenclature.

There are multiple ways to define neuronal cell classes (synaptic connections, electrophysiological properties, morphology, gene, or protein expression). Previous immunohistochemical investigations have already suggested that there are at least four classes of PG cells (Panzanelli et al., 2007; Parrish-Aungst et al., 2007). However, this classification has never been corroborated by distinctive functional properties and PG cell diversity is still most often overlooked.

In our last report, we proposed a classification with three subclasses of functionally distinct type 2 PG cells (Sanz Diez et al., 2019). We showed that a subclass of type 2 PG cells with prolonged ON-evoked responses that differ from those of previously described CR and Kv3.1-expressing PG cells also has a distinctive remarkably slow BF GABAergic IPSC. The present study further confirms that type 2.3 PG cells constitute a separate subclass by demonstrating its selective muscarinic input.

As requested by reviewer 1, I did additional experiments to determine whether the M1 mAChR is a molecular marker of this new subclass. The results suggest that this is not the case. M1 is widely expressed in the olfactory bulb in multiple cell types. This new result, combined with new recordings in granule cells, i.e. the most abundant OB interneurons, is nonetheless interesting as it suggests that despite the high expression level of M1 in multiple cell types, the synaptic pathway leading to the muscarinic activation of type 2.3 PG cells appears highly specific. The new data are presented in a new figure supplement (Figure 6 —figure supplement 1) and a new paragraph in the result section (page 14).

2. A better characterization of the BF neurons stimulated in the experiments with Dlx5/6-cre and Chat-cre mice. Reviewer #2 noted that some technical details regarding mice are missing, and that higher resolution images would, at minimum, be needed.

I apologize for this missing information. I have now included higher resolution images of BF fibers in the OB in figure supplements (Figure 2 —figure supplement 1 and Figure 3 —figure supplement 1). Each virus titre is now indicated (lines 732-735) and details on the Cre lines used (line 723-725) are provided in the methods.

All reviewers suggested a better characterization of transduced neurons: What fraction of Chat neurons are captured in dlx5/6 mice (e.g. assessed by IHC)?

We already characterized neurons labeled in BF-injected dlx5/6 mice using IHC (Sanz Diez et al., 2019). We reported that about 1/3 of the ChAT+ neurons were labeled. However, this quantification may not only reflect infection specificity but may also depend on infection rate and virus spread. If the question is on the nature of the ChAT+ neurons that were infected (those co-releasing ACh and GABA vs. those only releasing ACh), as reviewer 1 suggested, then IHC experiments do not respond to it. Our recordings indeed demonstrate that both types of cholinergic neurons are captured in BF-injected dlx5/6 mice. In addition to the exclusive cholinergic responses reported in the present manuscript, we also reported in Sanz Diez et al., (2019) mixed GABA/ACh responses in a subset of dSA cells recorded in the OB of BF-injected dlx5/6 mice (see an example in the response to reviewer 2). These responses are similar as those reported in Case et al., (2017) who demonstrated that they are mediated by a specific subpopulation of cholinergic neurons co-releasing ACh and GABA.

Do the Chat-negative neurons labeled in Chat-Cre mice project to OB (perhaps assessed with retrobeads from OB)?

As pointed out by reviewer 3, only 11% of the neurons labeled in ChAT mice were not co-labeled with ChAT in my IHC experiments suggesting a possible non-specific infection. In the other hand, the only type of response observed in my LCA recordings from >350 cells was the muscarinic excitation of type 2.3 PG cells. In addition, I recorded 54 PG cells in the whole-cell mode and only one of these cells responded with a possibly non-specific GABAergic response. This rare possibly unspecific response may have been mediated by a CB-, CR-, PV- or SOM-expressing BF GABAergic neuron. They all project to the OB, as revealed by our unpublished retrograde labeling experiments. However, this point is clearly not central to my study. Therefore, I have not done any additional experiments to determine the nature of these non-specifically labeled cells, which contribute little to the postsynaptic electrophysiological responses examined in this study.

Also, Dlx5/6-cre mice likely label many GABAergic neurons in the forebrain: Do the author manage to label only cells within the HDB? If not, where else may the GABAergic inputs come from and is it fair to unambiguously refer to them as BF inputs?

This is a fair concern, as it is likely that the virus infected a larger area than just the HDB/MCPO. Virus spread was not verified in injected mice from which the OB was extracted for electrophysiological experiments. However, our own retrograde tracing experiments (unpublished) together with those of Hanson et al. (2020), indicate that the other brain areas containing OB-projecting GABAergic neurons (i.e. the medial and lateral anterior olfactory nuclei, the lateral septum, the ventral subiculum) are all located quite far from our injection site in the BF making it unlikely that they contribute to the responses described in this study.

3. Reviewer#2 suggested that the "inhibition" of sTC and eTC by cholinergic stimulation be measured in a decrease in firing rate, as opposed to an increase in sIPCS frequency. It is presently unclear that the observed sIPSC effect comes from these cells. While a demonstration of this is likely beyond the scope of this study , showing that M1 receptors are selectively expressed in PG 2,3 cells (see point 1 above, see comments by reviewer #1) or that this effect is also dependent on M-channels would help bridge that gap.

I agree that the real impact of this new muscarinic pathway should be measured on the discharge of output neurons, which means not only on their firing rate but also on the synchrony of their discharge. This is a lot of work, beyond the scope of this study. As requested by reviewer 2, I replaced the term “inhibition” which implies a measure of firing activity in principal neurons with “IPSCs frequency” i.e. what was actually measured in my experiments. However, I do not agree that it is presently unclear that the increase in IPSCs frequency comes from type 2.3 PG cells. I indeed showed in Figure 7 and reported (page 17 lines 441-445) that “this response was blocked (n=8) or reduced (n=3) when the experiment was repeated in the presence of 2 µM pirenzepine (Figure 7C and 7E). Pirenzepine had little effect in 4/15 cells. A single photostimulation also transiently increased IPSC frequency in eTC (n=9) and s/mTC (n=8)(Figure 7B and 7D). Addition of pirenzepine attenuated this response in 10/11 cells.” This dependence on pirenzepine-sensitive M1 receptors together with the time course of the increase of IPSCs strongly support the idea that type 2.3 PG cells mediate the photo-evoked IPSCs in principal neurons.

4. Additional clarity on impact and a larger picture. Multiple things can/should be done in this regard:a. This paper follows another from the same group (Sanz Diez et al. 2019) that already looked at the postsynaptic effects of BF afferents onto OB cells using optogenetics in dlx5/6-cre mice. That paper focused mainly on GABAergic inputs, and reported dense innervation of all layers of the OB and potent GABAergic responses in granule cells, short axon cells and subpopulations of type 2 PG cells (but not type 1s). Reviewers feel that an analysis performed by the same group that ought to be repeated here to compare their new classification scheme with their previous one.b. This paper builds onto this 2019 paper, but focuses exclusively onto PG cells for both GABA and ACh responses. What motivates this focus is not presently clear, as several classes of OB neurons express nicotinic and muscarinic receptors (Castillo et al. 1999, Liu et al. 2015, Smith et al. 2015, Brunert and Rothermel 2021). A better justification for their exclusive focus on PG cells is warranted, together with some illustration that helps understand how cholinergic excitation and GABAergic inhibition of PG cells might affect OB output.c. Related to this, the focus of the paper remains vague: is it to define multiple subclasses of type 2 PG cells, in which case the functional GABAergic and cholinergic responses help substantiate the claim that these cells ought to be broken down into 3 subtypes? Or is the focus on the impact of BF inputs to OB function, in which case the focus on type 2 PG cells should be clarified (as well as the identity of the BF cells transduced with ChR2; see point 2)? Can we assume that ACh does not directly influence other cells? And what about the relative contribution of GABAergic signaling onto PG cells vs other interneurons and principal cells?

I thank the reviewing editor for these suggestions for improving the clarity and impact of my paper. In response to the many points raised in a- c, a new paragraph (page 3, lines 73-104, pasted below) and a new figure (Figure 1) have been added in the introduction. New Figure 1 includes a diagram of the OB glomerular microcircuits and a table that summarizes the properties of the different PG cell subtypes. The new paragraph clarifies the focus on PG cells (recent papers examined the BF connections on granule cells and the influences of this pathway on the OB circuits) and explains better their functions. It also introduces the new nomenclature proposed for the first time in this paper. I believe this addition disambiguates the definition of cell types and will help the reader contextualize the results:

*“*In such complex context, an important step towards understanding the influence and function of the BF inputs in the OB is to investigate the connections, temporal dynamics and functional impact of each BF pathway. […] Their output connections have not been determined.”

Concerning ACh influence on other cell types, new recordings in granule cells (mentioned page 14, lines 369-371) demonstrate that these abundant OB interneurons do not respond to the same stimulations that drive the muscarinic excitation of type 2.3 PG cells. These results further justify the focus on the muscarinic synaptic responses of type 2.3 PG cells, which is an unusual example of target-specific muscarinic transmission. This muscarinic pathway provides support for phasic, spatially restricted cholinergic transmission as opposed to spatially diffuse volume transmission. Yet, experiments are currently in progress in the lab using stronger pulse trains of light to determine whether volume transmission of ACh activates receptors in other cell types. However, these types of responses caused by diffusion of ACh ought to be less specific in terms of targets than synaptic transmission.

d. The utility of figure 1 is still unclear. All it does is to confuse the reader as to the potential existence of complex, co-transmission mechanisms, when in fact it derives from describing dlx5/6-cre mice as a specific driver of GABAergic neurons, which it isn't. Dlx5/6 labels neurons that originate in the embryonic ganglionic eminences during development, which includes GABAergic interneurons as well as cholinergic neurons in the forebrain. If one were interested in understanding how ACh modulates OB circuits, as the introduction suggests, one would not use dlx5/6-cre mice to start. Because the subsequent figures use more specific Cre lines, therefore, Figure 1 here may confuse readers who are not aware of the caveats associated with the Dlx5/6cre mice.

I recognize that starting with a figure summarizing the results obtained in dlx5/6 mice was an error. The new version of the manuscript directly starts with the description of muscarinic responses in ChAT-cre mice. Experiments in dlx5/6 mice come next and the specificity of this line which labels both GABA and ACh neurons is now clearly indicated (page 6-7, line 201). Previous figure 1, which shows the different types of response in dlx5/6-cre and their occurrence, is now shown in a modified version in Figure 3 —figure supplement 2.

Of note, viral injection in the BF of GAD2-cre (Bohm et al., 2020; Villar et al., 2021) or vGAT-cre mice (Hanson et al., 2020) produces the same axonal projection patterns in the OB as in dlx5/6-cre mice. Moreover, in these two models a significant fraction of the neurons labeled in the BF are ChAT-expressing neurons (up to 16% in GAD2-cre mice in Villar et al. (2021)). Co-labeling of cholinergic and GABAergic neurons is therefore not a specific caveat of the dlx5/6-Cre line.

Reviewer #1 (Recommendations for the authors):– The paper would be strengthened by converging approaches that add to the data presented. Specifically, M1 AChRs appear to be a molecular identifier of type 2.3 PG neurons. It would be interesting to perform in situ hybridization for M1 AChR to examine the distribution in the olfactory bulb to determine what extent it serves as a marker for this functionally identified class of neurons.

We have in the past characterized two functionally distinct classes of type 2 PG cells, CR-expressing cells (i.e. type 2.1, Benito et al., 2018) and Kv3.1-expressing cells (i.e. type 2.2, Najac et al., 2015). In our last report (Sanz Diez et al., 2019), we proposed the existence of a third class of type 2 PG cell (type 2.3) that has unique synaptic properties (prolonged ON-evoked response and prolonged BF IPSC) compared to the two previously described classes. The present study adds additional evidence that type 2.3 PG cells constitute a new subclass of PG cells by demonstrating its selective muscarinic input.

As requested by the reviewer, I have done new experiments to examine whether the M1 mAChR is a molecular marker for type 2.3 PG cells. I first tried to detect the expression of M1 mAChRs using an antibody from the Frontier Institute (Japan) that has been successfully used in the past (Yamasaki et al. JNeurosci 2010; Martinello et al., Neuron 2015). Unfortunately, I have not been able to label anything with this antibody (new batches of M1 antibody apparently do not work as well as older batches. Personal communication from the colleague who obtained the results in the Neuron 2015 paper). Then, I examined the distribution of an mRNA transcript of Chrm1, the gene encoding M1 mAChRs, using RNAscope fluorescence in situ hybridization. The results indicate that Chrm1 mRNA signals are widely expressed across all layers of the OB. The signal was particularly strong in granule cells. Thus, I did new LCA recordings in these interneurons indicating that they do not respond to the same photostimulation of the cholinergic afferents as those used to activate type 2.3 PG cells. I also combined FISH with immunohistochemistry to examine more closely the distribution of M1 in the glomerular layer. The results indicate that M1 receptors are non-selectively expressed by multiple cell types, including calretinin (CR) and tyrosine hydroxylase (TH) expressing cells. Thus, M1 is not a selective molecular marker of type 2.3 PG cells.

I have hesitated to include these new results in the paper because I am afraid that the first thing that comes to mind is that the labeling is unspecific. However, the data are consistent with a radiographic binding study (Le Jeune et al., 1995) but also with more recent ISH data in the Allen brain atlas (https://mouse.brain-map.org/experiment/show/73907497) which have been obtained with a different probe than the one provided for the RNAscope assay. Widespread expression of M1 in the OB is also confirmed by the immunohistochemical data shown in the human protein atlas (https://www.proteinatlas.org/ENSG00000168539-CHRM1/brain). Finally, preliminary analysis of available transcriptomic datasets also suggest that M1 is expressed by multiple cell types in the OB, including CR-expressing and CB-expressing neurons (Marcela Lipovsek, personal communication). Overall, these new data are therefore potentially interesting because they suggest that despite a widespread expression of M1 in the OB, only those on type 2.3 PG cells are activated by transient synaptic release of ACh. Thus, I have decided to include these data in a new figure supplement (Figure 6, figure supplement 1). They are described in the result section page 14.

– A novel finding of this paper is that a subset of BF neurons that release only ACh make direct connections with PG 2.3 neurons in the olfactory bulb. However, there is still a substantial population of neurons that co-release both ACh and GABA. In order to more fully appreciate the relative size of the population that releases ACh only, it would be helpful if the author could perform quantification of ChAT+ neurons in the BF that are not labeled by Dlx5/6. This could be achieved by performing ChAT immunohistochemistry in a Dlx5/6 cre mouse that was injected with a cre-dependent virus to estimate the relative size of the co-releasing and non-co-releasing populations.

If I understand well the reviewer, he/she suggests that a possible reason why dual GABA/ACh responses were not detected in my study may be that cholinergic neurons that co-release both transmitters are, for some reasons, not infected in the dlx5/6 mice. However, this is not the case. In our previous report (Sanz Diez et al., 2019), we estimated using immunohistochemistry that about 1/3 of the ChAT-expressing neurons were labeled in dlx5/6 mice that were injected with the same cre-dependent virus as the one used in the present study. We also reported that “in 3 of the 11 deep short axon cells tested, the response had two components that reversed at different potentials. At V_h_ = −70 mV, the response was biphasic with an outward component that was blocked by the GABAA receptor antagonist gabazine (GBZ, 5 μM), and an inward component that persisted in the presence of GBZ but was inhibited by the nicotinic receptor antagonist mecamylamine (20 μM)(not shown). These dual responses are similar to those mediated by basal forebrain neurons that release both GABA and acetylcholine onto a specific subtype of dSA cells (Case et al. 2017) and were therefore not further studied.” One example of these dual ACh/GABA response is shown in Author response image 1. These responses demonstrate that cholinergic neurons that co-release GABA and ACh are also labeled in the dlx5/6 mouse.

**Author response image 1. sa2fig1:** Light-evoked responses in BF-injected clx5/6 mice. Example of light-evoked responses recorded at two potentials in a dSA cell. At Vh=-20 mV, the response was dominated by an outward IPSC (top trace). At Vh=-70 mV, the outward current is reduced and followed by an outward component (middle). The outward component was totally blocked by GBZ (bottom). The EPSC that persisted in the presence of GBZ is mostly likely mediated by nicotinic receptors (See Case et el, 2017).

Reviewer #2 (Recommendations for the authors):Has the author tested the specificity of conditional expression using the AAV? The cre-dependence of conditional expression depends on the production method and the virus titre (Fisher et al. 2019, DOI: 10.1073/pnas.1915974116). If this is available, please describe this in a supplementary figure. If not, at the least, please include the titre information.

Those are fair concerns. The specificity of the AAV-mediated conditional expression in various types of GABAergic neurons as well as in ChAT-expressing neurons in the dlx5/6 mouse was verified in our previous report (Sanz Diez et al., 2019). I found minimal non-specific expression in the ChAT-cre mouse, as reported in the present study (lines 125-130, Figure 2B). Virus titer information are now indicated in the methods (lines 732-735).

Please include more information in the methods about the Cre lines used (stock number, original reference etc).

This information has now been added in the methods (lines 723-725).

It would be beneficial if the author could provide high resolution versions of the images used in Figure 1B and Figure 4C in the supplement, to be able to see the labelled structures better.

Done. A higher resolution image of BF axons distribution in the OB of ChAT mice is now shown in Figure 2—figure supplement 1. A higher resolution image of BF axons distribution in the OB of dlx5/6 mice is shown in Figure 3—figure supplement 1.

Figure 1C – if I understand correctly, the "excitation" in the pie chart is sometimes very brief type (as in Figure 1E), and sometimes the longer-lasting type, of the type analysed in Figure 3. It would be better if these could be labelled more precisely.

Figure 1 has now been deleted. A modified version appears as Figure 3—figure supplement 1. The labels of the pie chart have been modified. The 3 types of response are now called “brief excitation”, “inhibition only” and “mAChR excitation + inhibition”.

In general, the definition of cell types is ambiguous and not consistent with the definitions described elsewhere. For example, in Sanz Diez et al., the author used the response latency from the time of ON stimulation as a way to distinguish type 1 vs. type 2 PG cells. Please include analyses that allow comparisons with previous work(s). Similarly, a summary (e.g., a table) that describes the distinguishing properties of the PG cell types (types 2.1, 2.2 and 2.3) would help.

In this manuscript, I propose for the first time a new nomenclature of the different type 2 PG cell classes (types 2.1, 2.2 and 2.3). However, these different classes of type 2 PG cells were already identified in our previous study (Sanz Diez et al., 2019) and differentiated based on the same functional criteria as those used here. I apologize if this was not clear in the previous version. A new paragraph and a new Figure 1 has been added to the introduction in order to clarify the focus onto PG cells, help the reader contextualize the results, introduce this new nomenclature and disambiguate the definition of cell types. Figure 1 includes a diagram of the OB glomerular microcircuits as well as a table that describes the properties of the different PG cell subtypes. See the quote of this new paragraph in my response to the reviewing editor.

Concerning the classification as type 2 vs. type 1 PG cells, we found in our previous studies (Figure 3B in Najac et al., 2015; Figure 3E in Sanz Diez et al., 2019) that type 2 PG cells respond to the stimulation of the olfactory nerves with a plurisynaptic EPSC with an onset latency >2 ms whereas the monosynaptic response of type 1 PG cells has a shorter onset latency (<2 ms). An estimate of the onset latency of the ON-evoked response is now provided to support their classification as type 2 PG cells and allow comparison with our previous work. For instance, page 13 line 323 it is now indicated that “Evoked responses had an onset latency >2ms (average 3.14 ± 0.66 ms) as typically seen in pluri-synaptic ON-evoked responses of type 2 PG cells (Sanz Diez, Najac, and De Saint Jan 2019; Najac et al. 2015).” See also line 494 for type 2.2 PG cells. A zoom on the early part of the ON-evoked responses is also now shown in Figure 5F and in Figure 8D to illustrate onset latencies >2 ms.

Figure 7 – The effect on sTC and eTCs is an increase in iPSCs in voltage clamp, which is clearly shown. The term "inhibition", however, suggests a cessation of the action potential generation, which is not shown. As the net effect on the overall firing rates would likely depend on many factors, here, it is better to simply call it an increase in inhibitory synaptic inputs or similar.

This is correct. The term “inhibition” has been deleted and I now describe changes in inhibitory currents (IPSCs) frequency.

Reviewer #3 (Recommendations for the authors):– The presentation of the findings could be made stronger by providing a diagram of the OB and its component cells types and microcircuits. This would help non-experts understand which cell types are recorded from and how they fit into the shaping of odor responses in the OB. While the many subtypes of neurons populating the OB, their anatomical connections and functional roles remain incompletely understood, one issue with the current presentation of the findings is that we lose track of the forest for the trees. It is difficult to shake the feeling that we keep slicing smaller and smaller parts of a circuit that is not familiar to many, obscuring the significance of the findings. A better description of the roles that PG cells play in olfaction, especially vis-à-vis other interneurons and principal neurons, supplemented with a clear diagram, is necessary to help emphasize the importance of this work.

As indicated above in my responses to the reviewing editor and reviewer 2, a new paragraph and a new Figure 1 have been added to the introduction. The new figure shows a diagram of the OB glomerular microcircuits and a table that describes the properties of the different PG cell subtypes. I hope this addition clarifies the focus onto PG cells and the current knowledge about their function. I believe this addition will also help the non-expert readers contextualize the results and emphasize their significance. I thank the reviewer for the advices.

– The paper opens with a striking observation: stimulation of BF afferents evokes a range of excitatory and inhibitory responses in PG cells. However, as the paper unfolds and more specific methods targeting BF GABAergic and cholinergic neurons are leveraged, it becomes clear that the inhibitory and excitatory components originate in separate populations of afferents. The striking observation of Figure 1 is therefore nothing more than the result of non-selective axonal stimulation. Is starting with a confusing picture necessary, especially when more specific methods yielding more classical findings are utilized in later figures? Are there reasons to believe that BF GABAergic and cholinergic neurons are co-active in vivo?

I thank the reviewer for raising this problem with figure 1 that unnecessarily complicated the reading of the paper. Figure 1 has now been replaced and the Results section directly starts with the description of the target-specific muscarinic responses in ChAT-cre mice.

It is true that there is no reason to believe that the GABA and ACh separate inputs onto type 2.3 PG cells are co-active in vivo. It turns out that the inhibitory action on spiking of the GABAergic input (now shown in Figure 3 and 4) is much easier to see on top of a background muscarinic excitation.

– The molecular and synaptic basis for the segregation of GABAergic and cholinergic responses onto different subtypes of PG cells is not investigated. The author interprets this finding as resulting from differences in presynaptic innervation/specialization (e.g. lines 460, or 586-588). While probable, what evidence supports this possibility over others? Can we exclude postsynaptic components, such as differential postsynaptic clustering of GABA receptors, or expression of M1 receptors and/or M-channels? Can volume transmission can be excluded as a mode of release based on the data at hand?

The molecular and synaptic basis for the segregation of distinct GABAergic and cholinergic responses onto different subtypes of PG cells is outside the scope of this study. While different subunit compositions of the postsynaptic GABAA receptors undoubtedly determine the time course of BF IPSCs, we have also shown in our previous report (Sanz Diez et al., 2019) that BF GABAergic inputs have target-specific release properties (i.e. distinctive paired pulse ratio and failure rate) suggesting that they arise from distinct populations of neurons. This is now clearly indicated in the introduction (lines 65-66). Consistent with this, our unpublished retrograde labeling indicates that at least four non-overlapping populations of GABAergic BF neurons (labeled with calbindin, calretinin, somatostatin or parvalbumin) project to the OB. Also, in line with this idea, our experiments using transgenic lines expressing the Cre recombinase in specific GABAergic neurons show that distinct populations of GABAergic BF neurons innervate selective territories in the OB. For instance, CR-expressing BF neurons exclusively project in the granule cell layer whereas PV-expressing neurons project to the glomerular layer (a collaborative paper is in preparation on this finding). Thus, although postsynaptic mechanisms cannot be excluded, it is reasonable to think that different sub-populations of BF neurons modulate selective targets in the OB. There is a precedent in the literature: a specific sub-population of cholinergic neurons selectively innervates a sparse and specific subgroup of deep short axon cells located in the internal plexiform layer of the OB (Case et al., 2017). Identifying the populations of BF neurons that differentially modulate PG cell subclasses is the priority of our projects in the lab.

Concerning the cholinergic innervation, I focused on the muscarinic response of type 2.3 PG cells because it is evoked by a single brief stimulation of the afferent cholinergic fibers and it is target-specific. Activation of M1 receptors by a single transient of ACh suggests that postsynaptic M1 receptors are clustered near ACh release sites, which perhaps implies the existence of real anatomically-defined muscarinic synapses. This type of response is unusual in the brain. Slow M-channel-mediated muscarinic EPSP are usually evoked by strong and prolonged stimulations of the cholinergic neurons that favor spatial diffusion of ACh. The new FISH data suggest that expression of M1 is not specific to type 2.3 PG cells. Volume transmission of ACh may thus less specifically activate M1 receptors in several other cell subtypes. M2 receptors are also expressed in spontaneously active TH-expressing juxtaglomerular neurons. Their activation by bath-applied ACh inhibits spontaneous activity (Pignatelli et al., Chem senses 2008. doi:10.1093/chemse/bjm091). Experiments with stronger pulse-trains of ChR2 activation are ongoing in the lab to test whether they are activated by endogeneous ACh.

– Figure 9 provides evidence that GABA exerts a direct excitatory influence on some PG cells, possibly because their chloride reversal potential is elevated. The author should provide more evidence to substantiate this claim. For instance, does puffing GABA evoke a train of action potentials?

I think that further investigating this question would distract the reader from the main point. Here the principal result is not that GABA is depolarizing in some PG cells (this is consistent with the literature, see Parsa et al. PNAS 2015 or Smith and Jahr NatureNeurosci 2002), the main point is that GABA is depolarizing and excitatory in a *minority* of CR-expressing PG cells. The title (line 521) and conclusion of this paragraph (lines 552-55) have been slightly modified to emphasize this point.

We have shown that CR+ cells conserve functional properties of immature neurons (Benito et al., 2018). However, each cell is at a specific stage of maturation. An elevated chloride potential may be expected in the most immature ones. Puffing GABA would not evoke a train of APs. CR-expressing PG cells only express a single potassium current, the A-type. Once this current is inactivated, they cannot repolarize quickly. Thus, in response to prolonged depolarization, these cells only fire once and after the spike stay depolarized as shown in Pignatelli et al. (2016) and in Benito et al. (2018).

– Line 149: the author describes that AMPA, NMDA and nicotinic ACh receptor blockers prevent excitation of any interneurons. This is not true if one consideres some muscarinic receptors, and indeed this study goes on to show a striking example of this.

This is correct. Thanks for pointing out this error. The text (lines 147-149) has been modified as followed:

“Photo-evoked responses persisted in the presence of NBQX (10 µM), D-AP5 (50 µM) and mecamylamine (50 µM)(n=5, Figure 2E-F), which inhibit AMPA, NMDA and nicotinic ACh receptors, respectively. This cocktail of antagonist blocks a possible direct nicotinic excitation of PG cells (Castillo et al. 1999) as well as a putative indirect glutamatergic excitation following the nicotinic activation of mitral and tufted cells (Liu et al. 2015).”

– Line 169: remove 'always', since the rest of the sentence describes exceptions.

Done.

– Line 208-210: 89% of ChR2+ neurons in the BF of ChAT-Cre mice are cholinergic, leaving 11% of non-specifically labeled cells. Can the author provide any indication as to which other cells are labeled and whether they also project to OB, potentially contributing to postsynaptic responses?

The possibility that non-specifically labeled cell project to the OB was not verified as the electrophysiological recordings indicate that they contribute little if anything to the postsynaptic responses. See my detailed response to the reviewing editor.